# FRACTURED CHAIN-OF-THOUGHT REASONING

## ABSTRACT

Inference-time scaling techniques have significantly bolstered the reasoning capabilities of large language models (LLMs) by harnessing additional computational effort at inference without retraining. Similarly, Chain-of-Thought (CoT) prompting and its extension, Long CoT, improve accuracy by generating rich intermediate reasoning trajectories, but these approaches incur substantial token costs that impede their deployment in latency-sensitive settings. In this work, we first show that truncated CoT, which stops reasoning before completion and directly generates the final answer, often matches the full CoT sampling while using dramatically fewer tokens. Building on this insight, we introduce Fractured Sampling, a unified inference-time strategy that interpolates between full CoT and solution-only sampling along three orthogonal axes: (1) the number of reasoning trajectories, (2) the number of final solutions per trajectory, and (3) the depth at which reasoning traces are truncated. Through extensive experiments on five diverse reasoning benchmarks and several model scales, we demonstrate that Fractured Sampling consistently achieves superior accuracy-cost trade-offs, yielding steep log-linear scaling gains in Pass@k versus token budget. Our analysis reveals how to allocate computation across these dimensions to maximize performance, paving the way for more efficient and scalable LLM reasoning.[1]

## 1 INTRODUCTION

Recent advances in LLMs have enabled impressive capabilities in complex reasoning and problem solving (Guo et al., 2025; Kojima et al., 2022; Jaech et al., 2024; Brown et al., 2020; Hurst et al., 2024; Anthropic, 2024; Team et al., 2024). While much progress has been driven by scaling model size and training data (Hestness et al., 2017; Kaplan et al., 2020; Hoffmann et al., 2022), a complementary direction, *inference-time scaling*, has gained traction (Wang et al., 2023). This approach enhances performance by increasing computational effort at inference, without altering model parameters. Techniques such as self-consistency decoding (majority voting) (Wang et al., 2022), best-of-$n$ sampling (Stiennon et al., 2020; Brown et al., 2024; Cobbe et al., 2021; Dong et al., 2023), and ensemble-style methods (Yao et al., 2023; Zhou et al., 2022; Liao et al., 2025) leverage multiple forward passes to produce more accurate and robust predictions from instructed models.

In parallel with these inference-time scaling methods, another line of work has focused on improving the quality of individual reasoning paths. *Chain-of-Thought (CoT)* prompting (Wei et al., 2022) has emerged as a particularly effective technique by encouraging models to articulate intermediate reasoning steps before arriving at a final answer. Recently, *Long Chain-of-Thought (Long-CoT)* reasoning (Guo et al., 2025; Jaech et al., 2024) introduces longer and more diverse reasoning trajectories, often incorporating mechanisms like self-reflection and self-correction (Kumar et al., 2024). These extended CoTs explore a broader solution space and aggregate diverse intermediate steps into a single response. This has been shown to significantly improve accuracy and robustness, especially for tasks that require multi-step or logical reasoning. The downside is that they also dramatically increase token usage, resulting in higher inference costs.

Combining inference-time scaling with Long-CoT methods (e.g., using Long-CoT with self-consistency decoding) further amplifies this computational burden. Each technique alone may require thousands of additional tokens per input; together, they often push token budgets to impractical

---

[1]Code is available in the supplementary material.

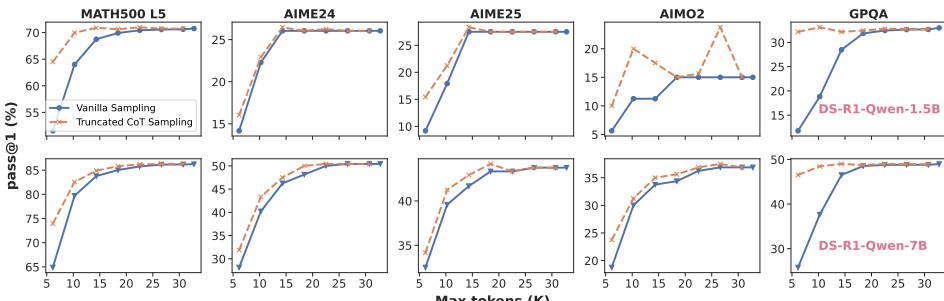

Figure 1: Pass@1 accuracy versus maximum token budget. Solid blue lines show the original full chain-of-thought (CoT) sampling, while dashed orange lines show our truncated CoT + response approach. Across all benchmarks, truncating the CoT (and generating the final answer) achieves equal or better accuracy with substantially fewer tokens, demonstrating that full CoT is unnecessary and that truncating CoT can save a large amount of computation without sacrificing performance.

levels, making such methods unsuitable for latency-sensitive or resource-constrained applications. This raises a central question:

*Can we retain the benefits of Long-CoT reasoning without incurring the full cost?*

To address this, we revisit the common assumption that complete Long-CoT traces are essential for accurate reasoning. Surprisingly, we find that *incomplete* CoT trajectories, i.e., traces truncated before the final answer, can still yield highly accurate results. As shown in Figure 1, across five reasoning benchmarks, simply truncating the CoT prefix and generating the answer (dashed orange) matches or even exceeds the accuracy of full CoT sampling (solid blue) given a max token constraint. This result challenges the notion that "more reasoning" always leads to better outcomes and suggests a new frontier for efficiency: *partial reasoning traces*.

To systematically trade off between cost and performance, we propose *Fractured Sampling*, a unified inference-time strategy that interpolates between full CoT and solution-only sampling. As illustrated in Figure 2(a), Fractured Sampling explores three orthogonal dimensions:

1. **Thinking trajectories:** the number of distinct CoT prefixes sampled;
2. **Solution diversity:** the number of final solutions generated per prefix;
3. **Thinking prefix length:** the depth at which each CoT is truncated.

Figure 2(b) further reveals that thinking steps dominate the overall token count, while final solutions contribute minimally, highlighting ample opportunities to optimize reasoning depth and breadth.

**Contributions.** Our key contributions are as follows: (1) We show that truncated CoT trajectories often achieve comparable or better performance than full CoT, at a fraction of the inference cost. (2) We propose *Fractured Sampling*, a unified inference-time framework that jointly controls reasoning depth, diversity, and token efficiency. (3) We provide a comprehensive analysis of the scaling behavior of Fractured Sampling across multiple reasoning benchmarks, offering practical insights into efficient inference strategies for LLMs.

## 2 PRELIMINARY

**Notations.** Let $x$ denote the input prompt and $\varepsilon$ be a random seed used to introduce stochasticity. The instruct LLM generates an initial response as follows: $z = f(x, \varepsilon)$, and a parser $g$ extracts the final answer: $y = g(z)$.

**Baseline inference techniques.** Before introducing our method, we review common sampling-based inference techniques widely used to enhance output quality from LLMs.

*Vanilla Sampling.* This approach generates $n$ independent completions with different random seeds:

$$\mathbb{F}_n(x, \varepsilon_{1:n}) = \{g \circ f(x, \varepsilon_i) \mid i = 1, \dots, n\}.$$

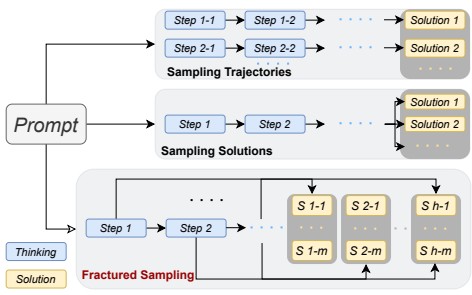 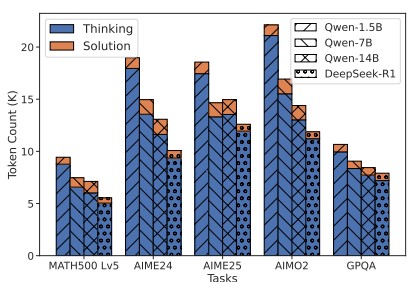

(a) Sampling strategies for reasoning LLMs.    (b) Token statistics.

Figure 2: (a) Comparison of sampling strategies for reasoning LLMs. Top: Sampling Trajectories–multiple complete reasoning chains are sampled independently from the model. Middle: Sampling Solutions–a single reasoning chain is used to generate diverse final solutions. Bottom: Fractured Sampling–our proposed method samples across both multiple reasoning trajectories and intermediate reasoning steps, enabling fine-grained control over diversity and computation. (b) Token statistics across tasks and models. Bars represent the average token count per sample, broken down into reasoning steps (blue) and final solutions (orange). The thinking process dominates the overall cost.

*Pass@k.* The pass@$k$ metric estimates the probability that at least one of the $k$ samples is correct:

$$\text{pass@}k = \mathbb{P}\left(\exists\, y_i \in F_k(x, \varepsilon_{1:k}) \text{ s.t. } y_i \text{ is correct}\right).$$

*Best-of-$n$.* This strategy selects the most confident response among $n$ candidates. If $s(z)$ denotes a scoring function (e.g., reward model), the best-of-$n$ output is:

$$y_{\text{best}} = g\left(\underset{z_i \in f(x, \varepsilon_{1:n})}{\arg\max}\; s(z_i)\right).$$

These inference-time strategies serve as foundations for reliability and robustness in model predictions, especially for reasoning-intensive tasks. Our approach builds on the sampling method by explicitly leveraging internal reasoning traces to enhance sample efficiency and answer diversity.

**Reasoning LLMs and long-CoT thinking process.** To better capture intermediate reasoning steps, reasoning-augmented LLMs use a CoT mechanism. Instead of producing a direct answer, the model first generates a reasoning trace:

$$h = [h_1^\varepsilon, \cdots, h_H^\varepsilon] = f_h(x, \varepsilon),$$

where $H$ denotes the total number of reasoning steps. The final solution is then generated conditioned on the full thought process:

$$z = f_o(x, h, \varepsilon).$$

This CoT formulation provides richer supervision and enables more structured sampling strategies, which our approach builds upon to enhance efficiency and performance.

To better reflect the internal reasoning process, we enhance diversity by sampling $m$ additional random seeds for each of $n$ thinking processes:

$$\mathbb{F}_{n,m}(x, \varepsilon_{1:n}, \varepsilon_{1:m}) = \left\{g \circ f_o(x, f_h(x, \varepsilon_i), \varepsilon_j)\,|\, i = 1, \cdots, n;\; j = 1, \cdots, m\right\}.$$

However, standard sampling methods only operate on the reasoning trajectory or the final solutions, overlooking the model's intermediate reasoning dynamics. *To fully exploit the internal structure of CoT reasoning, we propose sampling not just across independent trajectories, but also across intermediate reasoning steps.*

## 3 Fractured sampling for long chain-of-thought reasoning

To formalize the intermediate reasoning, the partial reasoning trace up to step $t$ is denoted as:

$$h_{1:t}^\varepsilon = [h_1^\varepsilon, \cdots, h_t^\varepsilon] = f_h^t(x, \varepsilon).$$

---

**Algorithm 1** 3D Sampling Framework (Full Trajectory → Segmentation → Solution Sampling)

---

**Require:** Prompt $x$; trajectories $n$; depth segments $H$; solutions per depth $m$; selector $\mathcal{S}$
**Ensure:** Final answer $\hat{y}$

1: $\mathcal{C} \leftarrow \emptyset$ ▷ All candidate answers
2: **for** $i = 1$ to $n$ **do**
3:     $T_i \leftarrow$ MODEL.GenerateFullCoT$(x)$
4:     Tokenize $T_i$ into $\{t_1, \ldots, t_L\}$
5:     $s \leftarrow \max(1, \lfloor L/H \rfloor)$ ▷ Segment size
6:     **for** $h = 1$ to $H$ **do**
7:         $p_{i,h} \leftarrow$ detokenize$(\{t_1, \ldots, t_{hs}\})$
8:         **for** $j = 1$ to $m$ **do**
9:             $\tilde{y}_{i,h,j} \leftarrow$ MODEL.GenerateSolution$(p_{i,h}, x)$
10:           $a_{i,h,j} \leftarrow$ ExtractAnswer$(\tilde{y}_{i,h,j})$
11:           $\mathcal{C} \leftarrow \mathcal{C} \cup \{a_{i,h,j}\}$
12:         **end for**
13:     **end for**
14: **end for**
15: $\hat{y} \leftarrow \mathcal{S}(\mathcal{C})$
16: **return** $\hat{y}$

---

Our approach leverages intermediate reasoning traces to aggregate predictions, thereby enhancing both efficiency and diversity. The key idea is to decompose the response generation into multiple stages and perform aggregation not only over independent final responses but also across intermediate reasoning steps.

**Fractured sampling.** Fractured sampling extends this idea by incorporating intermediate reasoning stages directly into the sampling process. Specifically, we sample solutions at each step of the reasoning chain:

$$\mathbb{F}_{n,m,H}(x, \varepsilon_{1:n}, \varepsilon_{1:m,1:H}) = \left\{ g \circ f_o \left( x, f_h^t(x, \varepsilon_i), \varepsilon_{j,t} \right) \big| i = 1, \cdots, n; \; j = 1, \cdots, m; \; t = 1, \cdots, H \right\}.$$

Here, $f_h^t(x, \varepsilon_i)$ denotes the partial reasoning trace up to step $t$, and $\varepsilon_{j,t}$ is the random seed used for generating the response at that stage. By aggregating responses across all $H$ intermediate steps, fractured sampling captures the evolving thought process and synthesizes diverse insights into a more robust final answer.

Fractured sampling offers two primary advantages: (1) *Granular Aggregation:* Integrating intermediate reasoning steps enables early detection of conclusions and avoid overthinking, improving the consistency of final predictions. (2) *Enhanced Diversity:* The multi-level sampling mechanism encourages a wide range of reasoning trajectories. Aggregating these paths produces a consensus that is more resilient to individual failures.

**Three orthogonal dimensions of sampling.** As shown in Algorithm 1, Fractured Sampling unifies and extends existing sampling strategies by operating along three orthogonal axes:

- $m$: Solution Diversity — sampling multiple final outputs from a single reasoning trace.

- $n$: Trajectory Diversity — sampling multiple independent reasoning traces with different seeds (vanilla CoT sampling).

- $H$: Reasoning Depth Diversity — sampling at different intermediate stages of a single reasoning trace (unique to fractured sampling).

This tri-dimensional framework enables a fine-grained exploration of the cost–performance landscape. While $m$ and $n$ offer diversity at the output or full-trajectory level, the $H$ dimension uniquely captures the temporal evolution of reasoning, offering early, diverse, and efficient decision points. Together, they provide a powerful toolkit for scalable and reliable inference-time reasoning.

Empirical results (see Section 4) show that fractured sampling is a strong methods to produce diverse and meaningful solutions.

## 3.1 ANALYSIS OF FRACTURED SAMPLING

**Fractured sampling benefits from diverse solutions.** By distributing samples across both trajectories and intermediate steps, fractured sampling capitalizes on diverse error modes to boost overall success. The following proposition provides an analysis about our phenomenon.

**Proposition 1** (Diversity Lower Bound, informal). *Let $F_k$ be the indicator of failure for branch sample $k$ and $q_k = P(F_k = 1)$. Then the fractured-sampling success probability satisfies*

$$p_{\text{seg}} = 1 - \Pr\big(\wedge_{k=1}^{K} F_k = 1\big) = 1 - \mathbb{E}\Big[\prod_{k=1}^{K} F_k\Big],$$

*and by inclusion–exclusion*

$$\mathbb{E}\Big[\prod_{k=1}^{K} F_k\Big] = \prod_{k=1}^{K} q_k + \sum_{i<j} \text{Cov}(F_i, F_j) + \cdots.$$

That is to say, negative covariance $\text{Cov}(F_i, F_j) \leq 0$ means that failures at two different samples $i, j$ tend not to coincide, i.e. the two sampling locations provide *diverse* error modes. If we only consider the second order expansion, we have $p_{\text{seg}} \geq 1 - \prod_{k=1}^{K} q_k = 1 - \prod_{t=1}^{H}(1 - p_t)^m$. Fractured sampling spreads samples across intermediate steps to maximize this diversity: because failures are unlikely to all happen together, the probability that *every* sample fails is strictly less than the naïve product of their marginal failure rates. Consequently, the overall success probability $p_{\text{seg}}$ is boosted above the independent-baseline $1 - \prod(1 - p_t)^m$.

To understand the limits of fractured sampling, we examine two extreme correlation regimes among the $K = mH$ branch samples.

**Almost perfect correlation.** When every sample fails or succeeds in unison ($F_i = F_j$ almost surely), the entire set of $K$ trials collapses to a single Bernoulli event. In this case,

$$\Pr(F_1 = \cdots = F_K = 1) = q, \qquad p_{\text{seg}} = 1 - q,$$

so the sampling reduces to plain single-step sampling and yields no extra benefit. Sampling only along the $m$-axis (multiple outputs per trace) behaves similar to this.

**Full independence.** If all $F_k$ are mutually independent with $\Pr(F_k = 1) = q_k$, then

$$\Pr(F_1 = \cdots = F_K = 1) = \prod_{k=1}^{K} q_k = \prod_{t=1}^{H}(1 - p_t)^m, \quad p_{\text{seg}} = 1 - \prod_{t=1}^{H}(1 - p_t)^m.$$

The sampling achieves the product-of-marginals bound: diversity arises purely from geometric averaging of each step's success rate. Standard trajectory sampling ($n$-axis) behaves similar to this regime with a single successful rate $p$.

**Intermediate regimes.** Between these extremes, negative pairwise covariances ($\text{Cov}(F_i, F_j) < 0$) drive the all-fail probability below the independent baseline, delivering gains beyond simple marginal aggregation. By contrast, positive correlations ($\text{Cov}(F_i, F_j) > 0$) erode this advantage, interpolating smoothly between full independence and perfect correlation. Sampling along the depth dimension $H$ exploits these intermediate correlations to maximize diversity and overall success.

As illustrated in Figure 3, the correlation matrices shows how failure events at different reasoning depths $H$ co-occur across five diverse benchmarks. Dark green cells along the diagonal indicate that failures at the same depth are, by definition, almost perfectly correlated. More interestingly, the off-diagonal pattern varies by task: many entries are light or even pink (negative), signalling that failures at two distinct depths tend not to happen simultaneously. This negative covariance across depths is precisely what fractured sampling exploits, by spreading samples over intermediate stages, it decorrelates error modes and thus markedly reduces the probability that all branch samples fail together. Benchmarks with stronger negative off-diagonal structure (e.g. GPQA) exhibit the largest gains from fractured sampling, confirming our theoretical diversity lower-bound analysis.

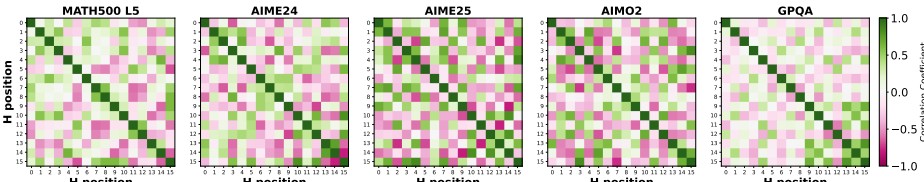

Figure 3: Correlation matrices of binary failure indicators across intermediate reasoning depths (positions $H$) under fractured sampling for five benchmarks. Each cell shows the Pearson correlation coefficient between failure events at two depth positions; green denotes positively correlated failures (synchronized error modes), while pink denotes negatively correlated failures (diverse error modes) that fractured sampling exploits to boost overall success.

## 3.2 SCALING LAWS ALONG THE TRAJECTORY DIMENSION

In fractured sampling, we allocate computation across three orthogonal axes $(n, m, H)$. Here, we hold the branching factor $m$ and fracturing depth $H$ constant, and investigate how increasing the number of independent trajectories $n$ affects performance under a fixed token budget. Denote the total tokens consumed as

$$B(n, m, H) = n\,C_{\text{thinking}} + n\,m\,H\,C_{\text{solution}} = n\big[C_{\text{thinking}} + mH\,C_{\text{solution}}\big],$$

where $C_{\text{thinking}}$ is the average tokens per trajectory spent on "thinking" (the reasoning prefix), and $C_{\text{solution}}$ is the per-step cost of generating each candidate solution.

**Log-linear scaling behavior.** Empirical studies across diverse benchmarks reveal a remarkably consistent log-linear relationship between computational budget and success rate along each axis:

$$\text{pass@}k\big(B_n\big) \approx C_n \log B_n + c_n, \qquad B_n = B(n, 1, 1),$$
$$\text{pass@}k\big(B_m\big) \approx C_m \log B_m + c_m, \qquad B_m = B(1, m, 1),$$
$$\text{pass@}k\big(B_H\big) \approx C_H \log B_H + c_H, \qquad B_H = B(1, 1, H).$$

Here, the constants $C_n, C_m, C_H$ measure the marginal gain in log-budget per unit improvement in pass rate, while $c_n, c_m, c_H$ capture dataset-specific offsets.

**Depth yields the steepest slope.** Across a range of tasks, we consistently find

$$C_H \geq \max\{C_n, C_m\},$$

indicating that allocating tokens to deeper intermediate sampling (the $H$ axis) produces the largest incremental improvements per token. Intuitively, early-stage branching captures coarse but high-signal glimpses of the solution space, allowing the model to "course-correct" before committing to full trajectories and thus yielding higher gains for each additional intermediate sample.

**Beyond single-axis scaling.** While single-axis laws offer valuable intuition, actual performance often improves when $(n, m, H)$ are tuned jointly. Since the $n$-axis contributes additively and independently, we condition on fixed $(m, H)$ and model

$$\text{pass@}k\big(B\big) \approx C_{m,H} \log B\big(n \mid m, H\big) + c_{m,H},$$

where the coefficient $C_{m,H}$ encapsulates the combined effect of branching factor and depth. These cross-terms reveal synergistic gains or trade-offs between axes, guiding more nuanced budget allocations. We explore these interactions and derive dataset-specific strategies in Section 4.

## 4 EMPIRICAL RESULTS

**Settings.** All inference experiments are conducted using NVIDIA A100-80GB GPUs, leveraging the vLLM framework (Kwon et al., 2023). Following the sampling configuration recommended by Guo et al. (2025), we set `temperature=0.6`, `top_p=0.95`, and `max_tokens=32768`. Our primary focus is on models from the DeepSeek-R1 family (Guo et al., 2025), and we further

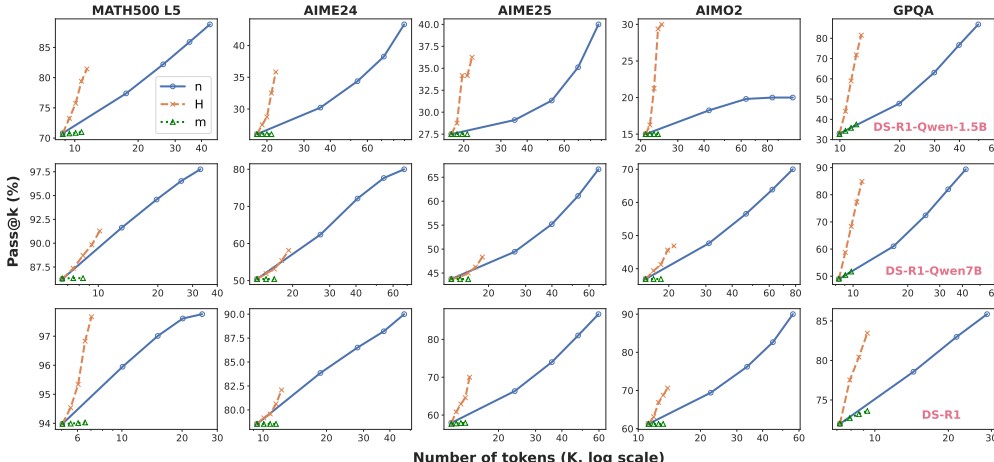

Figure 4: Pass@$k$ performance versus token budget ($log_{10}$). We compare: **m**–sampling only the final solution; **n**–sampling full reasoning trajectories; **H**–fractured sampling across all intermediate steps. Fractured sampling consistently yields higher pass@$k$ at a given token budget. Refer to Figure E.1 and E.2 for DeepScaleR, Qwen3 and GPT-OSS with a similar pattern.

validate our findings using reasoning models from Qwen3 (Team, 2025), Skywork-OR1 (He et al., 2025), DeepScaler (Luo et al., 2025) and GPT-OSS (Agarwal et al., 2025). For clarity, we refer to DeepSeek-R1 as DS-R1, DeepSeek-R1-Distill-Qwen-1.5B as DS-R1-Qwen-1.5B, DeepScalerR-1.5B-Preview as DSR-1.5B, and Skywork-OR1-7B as SW-OR1-7B.

Evaluation is performed on five challenging math and scientific reasoning benchmarks: MATH500 Level 5 (L5) (Lightman et al., 2023), AIME24, AIME25 (MAA Committees, 2025), AIMO2 reference questions (Frieder et al., 2024), and the GPQA Diamond set (Rein et al., 2024). Unless otherwise specified, we set $n = 16$, $H = 16$ and $m = 4$. $H = 16$ indicates that the original thinking CoT is divided into 16 equally sized segments based on token count. For instance, the third fractured CoT consists of the first three segments of the full thinking trajectory.

## 4.1 Scaling Law for Each Dimension

Figure 4 plots $\text{pass@}k$ versus total tokens $B$ for three sampling dimensions. Across all benchmarks and models, fractured sampling exhibits the steepest log-linear gains per token. In particular, we fit

$$\text{pass@}k\big(B_*\big) \approx C_* \log B_* + c_*, \quad * \in \{n, m, H\},$$

and consistently observe $C_H \geq \max\{C_n, C_m\}$. This confirms that allocating budget to intermediate-step branching yields higher marginal returns than either sampling more independent traces or more final solutions alone.

**Interpreting the gains.** Fractured sampling captures rich, underutilized variation in intermediate reasoning states, allowing the model to "course-correct" early and avoid committing to error-prone trajectories. This leads to: (1) *Higher early returns:* At small budgets, $H$–sampling yields a much steeper rise in pass rate than $n$ or $m$, since few intermediate samples can quickly pinpoint correct partial reasoning. (2) *Consistent dominance:* The gap between the $H$-curve and the others persists across all budgets, demonstrating robustness to scale. (3) *Task-dependent effect:* Benchmarks with less positive error correlations across depths (e.g. GPQA) show the largest absolute improvement from fractured sampling, in line with our diversity-bound analysis.

These results empirically validate that, under the same compute budget, fractured sampling shifts the inference-time scaling curve upward, *achieving higher accuracy at lower cost* by leveraging the temporal structure of chain-of-thought.

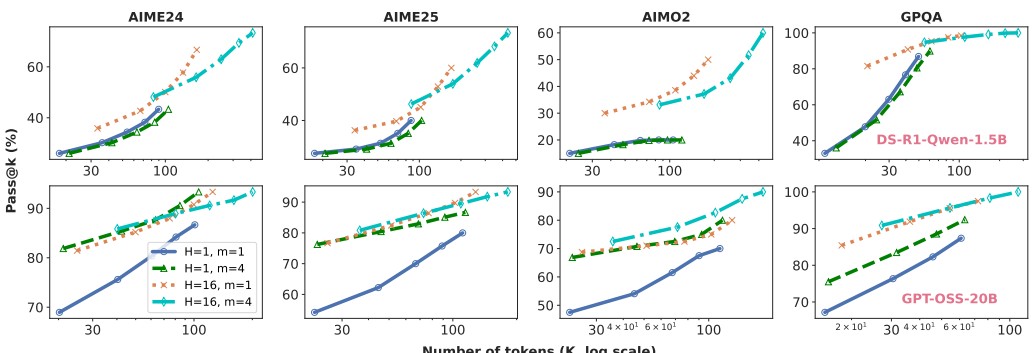

Figure 5: Pass@$k$ performance versus total token budget ($log_{10}$) for four sampling schemes. In each subplot, we compare: **H=1, m=1**–only sampling full reasoning trajectory; **H=1, m=4**–sampling both full reasoning trajectories and the final solution; **H=16, m=1**–both full reasoning trajectory sampling and fractured sampling across all intermediate steps; **H=16, m=4**–sampling all three dimensions. $n$ is in [1, 2, 4, 8, 16] for the five points (from left to right) on each line ($n \leq 8$ on GPQA). Refer to Figure E.3 for DeepScaleR and Qwen3 with a similar pattern.

## 4.2 SCALING LAW ACROSS DIMENSIONS

Thus far we have examined each sampling axis in isolation. Figure 5 extends this analysis by comparing four representative schemes that allocate budget across the solution ($m$) and depth ($H$) dimensions simultaneously, with the trajectory axis ($n$) swept to 16. We have: (1) ($H$=1, $m$=1): standard single-path CoT sampling (baseline). (2)($H$=1, $m$=4): augment baseline with 4 final answers per trajectory. (3) ($H$=16, $m$=1): fractured sampling across 16 depths, one answer each. (4) ($H$=16, $m$=4): full three-axis sampling (*both* deep fracturing and multiple final answers).

Across every task and model, non-baseline schemes (excl. ($H$=1, $m$=4)) outperform ($H$=1, $m$=1) at fixed budget. More importantly, expanding $H$ is usually more effective than expanding $m$. These multi-axis scaling laws reveal that, under the same token budget, the most efficient use of compute is to allocate tokens for temporal branches $H$. Final-solution replicates $m$ may also work in some cases, especially for GPT-OSS (see Figure E.2 and 5).

With such a high pass@k from a low token budget, we believe our unified sampling across three dimensions paves a new way for reinforcement learning (RL), where both high pass@k and efficient sampling are important for the large-scale RL training. We leave this exploration to future work.

## 4.3 ACCURACY ACROSS DIMENSIONS

In prior experiments, we reported the metric pass@k, which indicates whether a correct prediction is present among a set of generated samples. In this section, we further examine whether a correct solution can be identified by (1) a reward model from the predictions generated across the three sampling axes; or (2) majority voting. We employ the process reward model (PRM), specifically Qwen2.5-Math-PRM-72B (Zhang et al., 2025), which has shown strong performance across a range of PRM benchmarks. Due to its limited context window (4K tokens), we score only the final solution, rather than the intermediate reasoning steps. The reward assigned to the final step is used as the overall score for the entire solution.

As shown in Table 1, sampling with $H = 1, m = 4$ yields a modest improvement in average accuracy compared to the standard sampling setting of $H = 1, m = 1$ (61.6% vs. 60.4% for BoN and 66.8% vs 66.7% for Maj). Interestingly, increasing only the $H$ dimension to $H = 16, m = 1$ leads to a slight improvement for BoN and degradation for Maj, which contrasts with our earlier observation that varying $H$ is typically more effective than varying $m$ in terms of pass@k. We hypothesize that incorporating all $H = 16$ generated solutions introduces excessive noise, making it challenging for a PRM to correctly identify the optimal solution. This may be due to two factors: (1) the Long-CoT model tends to generate coherent and logically consistent solutions, which are difficult for the PRM to differentiate; (2) the PRM is trained predominantly on simpler and short-

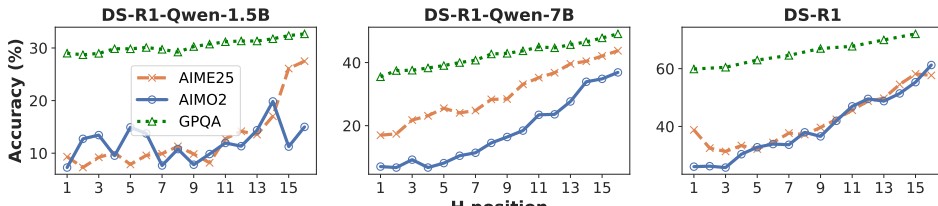

Figure 6: Accuracy versus the position of Fractured CoT. We split the whole reasoning trajectory into 16 intermediate steps equally, and observe: (1) Even with a $\frac{1}{16}$ reasoning trajectory, the accuracy is still decent, especially for GPQA; (2) More reasoning tokens lead to higher accuracy.

CoT data and may struggle to evaluate responses to more complex and long ones. More noise also causes a worse Maj accuracy.

Motivated by the trend observed in Figure 6—where later reasoning positions (i.e., higher $H$ indices) are associated with improved accuracy—we apply a simple denoising strategy by discarding earlier solutions ($H = 1$ to $H = 11$) and retaining only the last four ($H = -4$). This simple adjustment significantly enhances performance, raising the accuracy from 61.4% ($H = 16, m = 1$) to 68.0% ($H = -4, m = 1$) for BoN and 65.3% to 69.9% for Maj. Further combining both dimensions

Table 1: Best-of-N and majority voting accuracy with different dimensional settings. $n = 16$ here for all settings. $H = 1, m = 1$ denotes the standard sampling setting. $H = -4$ here means that we take the last 4 solutions among all 16 predictions in the $H$ dimension.

| Metric | H | m | MATH500 L5 | AIME24 | AIME25 | AIMO2 | GPQA | Avg. |
|--------|---|---|------------|--------|--------|-------|------|------|
| BoN | *DS-R1-Qwen-7B* | | | | | | | |
| | 1 | 1 | 90.3 | 63.3 | 53.3 | 40.0 | 55.1 | 60.4 |
| | 16 | 1 | 90.3 | 70.0 | 53.3 | 40.0 | 53.5 | 61.4 |
| | -4 | 1 | 93.3 | **73.3** | **60.0** | 60.0 | 53.5 | 68.0 |
| | 1 | 4 | 90.3 | 70.0 | 53.3 | 40.0 | 54.6 | 61.6 |
| | 16 | 4 | 92.5 | **73.3** | **60.0** | 50.0 | **57.6** | 66.7 |
| | -4 | 4 | **94.8** | **73.3** | **60.0** | **70.0** | 56.1 | **70.8** |
| | *DS-R1-Qwen-14B* | | | | | | | |
| | 1 | 1 | 91.8 | 80.0 | 60.0 | 50.0 | 59.6 | 68.3 |
| Maj | *DS-R1-Qwen-7B* | | | | | | | |
| | 1 | 1 | 95.5 | 76.7 | 60.0 | 50.0 | 51.5 | 66.7 |
| | 16 | 1 | 94.0 | 73.3 | 60.0 | 50.0 | 49.0 | 65.3 |
| | -4 | 1 | 96.3 | 76.7 | 63.3 | **60.0** | 53.0 | 69.9 |
| | 1 | 4 | 95.5 | 76.7 | 60.0 | 50.0 | 52.0 | 66.8 |
| | -4 | 4 | **96.3** | **80.0** | **66.7** | **60.0** | **53.5** | **71.3** |
| | *DS-R1-Qwen-14B* | | | | | | | |
| | 1 | 1 | 94.0 | 83.3 | 53.3 | 50.0 | 59.6 | 68.0 |

($H = -4, m = 4$) yields an accuracy of 70.8% for BoN and 71.3% for Maj, a 10.4% and 4.6% improvement over the baseline setting ($H = 1, m = 1$). Notably, this configuration even outperforms standard sampling with a larger model that has twice the number of parameters (70.8% vs. 68.3% for BoN, and 71.3% vs 68.0% for Maj).

### 4.4 EARLY STOPPING FOR EFFICIENT GENERATION

From the perspective of inference efficiency, we explore whether the consistency of predictions across the $H$ dimension can be leveraged for early stopping. Specifically, if a particular prediction appears with high frequency (i.e., exceeds a predefined threshold) across multiple $H$ positions, we consider this as a signal to terminate the generation early, thereby reducing computational cost.

As illustrated in Figure 6, prediction accuracy tends to be low at earlier positions. When the reasoning trace is divided into too many intermediate steps (i.e., a larger $H$), the model must generate a correspondingly large number of partial solutions, each requiring additional tokens. To balance computational efficiency and accuracy, we empirically initialize the first $H$ position at a token index of 6144 and evaluate predictions at every subsequent 2048-token interval. For example, given a question, the model first generates 6144 reasoning tokens. Based on these tokens, a solution is generated and a prediction is extracted. Then, conditioned on the original question and the previously generated 6144 reasoning tokens, the model continues generating another 2048 tokens to produce the next prediction. Generation terminates once the same prediction occurs more than once or when the maximum token limit (`max_tokens`) is reached. In the latter case, we adopt the final prediction, as later predictions tend to benefit from more extensive reasoning.

Table 2: The relative performance for early stop compared to vanilla sampling. Early stopping significantly saves inference budget ($\approx 20\%$) while preserving or outperforming vanilla's accuracy.

| Model | Method | Accuracy (%) ↑ | | | | | Number of Tokens per Question (K) ↓ | | | | |
|---|---|---|---|---|---|---|---|---|---|---|---|
| | | MATH500 | AIME25 | AIMO2 | GPQA | Avg. | MATH500 | AIME25 | AIMO2 | GPQA | Avg. |
| DS-R1-1.5B | Vanilla | 70.8 | 27.5 | 15.0 | 34.1 | 36.9 | 8.8 | 17.4 | 21.1 | 10.0 | 14.3 |
| | Early Stop | **+1.2** | **-0.0** | **+10.6** | -0.3 | **+2.9** | **-1.4** | **-2.3** | **-7.1** | **-0.6** | **-2.9** |
| DSR-1.5B | Vanilla | 76.5 | 41.7 | 20.0 | 19.2 | 39.4 | 5.1 | 8.3 | 10.6 | 7.8 | 8.0 |
| | Early Stop | -0.9 | **-0.0** | **-0.0** | **+0.4** | -0.1 | **-0.9** | **-1.1** | **-3.6** | **-0.2** | **-1.5** |
| SW-OR1-7B | Vanilla | 89.0 | 45.0 | 47.5 | 48.6 | 57.5 | 6.7 | 13.4 | 14.9 | 8.5 | 10.9 |
| | Early Stop | -0.4 | **-0.0** | **-0.0** | **+1.1** | **+0.2** | **-0.6** | **-2.5** | **-3.3** | **-2.2** | **-2.2** |

As shown in Table 2, this early stopping strategy preserves accuracy and, in some cases, improves it—achieving a 2.9% increase for DeepScaleR-1.5B-Preview. In terms of computational efficiency, early stopping reduces the number of generated tokens by approximately 20% compared to standard generation. Notably, this method is simple to implement and requires no additional training.

**Related Work.** Due to the page limit, please refer to Appendix A.

## 5 CONCLUSION

In this work, we introduce Fractured Sampling, a new Long-CoT inference paradigm that seamlessly unifies partial-trace and final-answer sampling by jointly controlling reasoning depth, trajectory diversity, and solution diversity. We uncover consistent log–linear scaling trends along each axis and offer theoretical insights into how sampling across intermediate reasoning steps maximizes diversity and per-token gains. Fractured Sampling redefines the cost–performance frontier of chain-of-thought inference, enabling powerful reasoning in LLMs with lower computational overhead.

## ETHICS STATEMENT

Our work introduces fractured sampling, an inference-time strategy that improves the cost-effectiveness of CoT reasoning in LLMs. By enabling finer-grained control over reasoning depth and sampling diversity, our method offers a principled way to reduce token usage while maintaining or improving performance on complex reasoning tasks. This has several potential societal and technological implications.

On the positive side, reducing inference costs can improve the accessibility and environmental sustainability of LLM-based systems, especially in domains like education, healthcare, and scientific research where computational resources may be limited. fractured sampling could enable more efficient deployment of advanced reasoning capabilities on edge devices or in latency-sensitive applications.

However, our approach assumes access to and control over the internal sampling process of LLMs, which may not be feasible in all commercial or black-box settings. Broader adoption of these techniques will require transparency and responsible integration within larger model-serving pipelines.

We hope our work encourages further research into efficient, reliable, and interpretable reasoning strategies in LLMs, with careful attention to the ethical and societal impacts of their deployment.

## REPRODUCIBILITY STATEMENT

We attached our code for the main results shown in this paper in the supplementary material. In addition, our experimental settings (benchmarks, models and hyperparameters) are clearly stated in Section 4.

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

## A    RELATED WORK

**Test-time scaling law.** Scaling laws have traditionally described how model performance improves with increased training compute (Hestness et al., 2017; Kaplan et al., 2020; Hoffmann et al., 2022), e.g., through more supervised fine-tuning or reinforcement learning steps. However, a complementary class of *test-time scaling laws* has emerged (Snell et al., 2024; Jaech et al., 2024), which characterizes performance gains obtained purely by increasing inference-time budget, without modifying model parameters. This includes techniques such as self-consistency decoding (Wang et al., 2022), best-of-$n$ sampling (Brown et al., 2024; Cobbe et al., 2021; Dong et al., 2023). On the other hand, CoT prompting, where performance improves with more samples or longer reasoning traces (Wei et al., 2022). Recent work, including the O1 and R1 series (Jaech et al., 2024; Guo et al., 2025), further demonstrates that extended trajectories (e.g., Long CoT) with multiple rollouts yields predictable improvements under test-time scaling curves.

On the other hand, Process Reward Models (PRMs) (Lightman et al., 2023; Zhang et al., 2024a; Wang et al., 2023) further enable fine-grained control by assigning dense, step-level rewards, which can guide search methods like Monte Carlo Tree Search (Luo et al., 2024). However, most approaches scale only along coarse dimensions, such as sample count or token length, or require external supervision via PRMs for finer control. In this work, we propose a more fine-grained view through *Fractured Sampling* without relying on PRMs, which explicitly decomposes generation into multi-stage reasoning traces and enables aggregation at intermediate steps. This design reveals richer scaling behaviors across trajectory depth, diversity, and stage-wise composition, and offers a more nuanced understanding of inference-time compute allocation.

**Efficient sampling for LLMs.** As large language models grow in size and capability, their inference cost becomes a significant bottleneck (Wan et al., 2023), especially when relying on multi-sample or multi-turn decoding strategies in reinforcement learning (Ouyang et al., 2022; Xiong et al., 2023; Dong et al., 2024; Xiong et al., 2025; Shao et al., 2024) or large-scale serving (Ainslie et al., 2023). This has motivated a line of work on *efficient sampling*, which aims to reduce compute without sacrificing performance. Approaches such as speculative decoding (Stern et al., 2018; Leviathan et al., 2023; Xia et al., 2024; Chen et al., 2023a; Zhang et al., 2023; Sun et al., 2024; Chen et al., 2023b; Li et al., 2024b; Liao et al., 2025), KV cache pruning (Xu et al., 2024; Xiao et al., 2023; Zhang et al., 2024c; Li et al., 2024a; Ge et al., 2023; Zhang et al., 2024b; Yang et al., 2024; Liu et al., 2024), are widely used in real-world LLM services. While these methods achieve notable efficiency gains, they largely operate within a fixed test-time scaling curve: improving the efficiency of a given point on the curve without fundamentally changing its shape. In contrast, we argue that the most principled path forward lies in reshaping the scaling law itself: by rethinking how inference budget is allocated across reasoning stages and sampling axes, one can unlock qualitatively different compute-performance tradeoffs. Our proposed *Fractured Sampling* method embodies this principle, revealing richer scaling dynamics and enabling more cost-effective reasoning through staged aggregation.

## B    LLM USAGE

In this paper, we mainly use LLM to rephrase sentences, making them more grammar-correct and fluent. We don't apply it for other usage.

## C    LIMITATIONS

While Fractured Sampling demonstrates significant improvements in the efficiency and performance of CoT reasoning, the effectiveness of our approach depends on access to a long and coherent reasoning trajectory from the model. Tasks or models that do not naturally exhibit such structured intermediate reasoning may benefit less from intermediate-stage sampling.

## D    PROOF OF THE DIVERSITY LOWER BOUND

*Proof.* Let $q_i = \Pr(F_i = 1) = \mathbb{E}[F_i]$ and denote

$$\mu_{ij} = \mathbb{E}[F_i F_j], \qquad \mu_{ijk} = \mathbb{E}[F_i F_j F_k], \qquad \mu_{ijkl} = \mathbb{E}[F_i F_j F_k F_\ell].$$

Define the joint cumulants

$$\kappa_{ij} = \mu_{ij} - q_i q_j,$$
$$\kappa_{ijk} = \mu_{ijk} - \mu_{ij} q_k - \mu_{ik} q_j - \mu_{jk} q_i + 2 q_i q_j q_k.$$

Using the inclusion-exclusion identity $\prod_k F_k = \sum_{I \subseteq [K]} (-1)^{|I|} \prod_{i \in I} (1 - F_i)$ and collecting equal-order terms yields the exact expansion:

$$\mathbb{E}\left[\prod_{k=1}^{K} F_k\right] = \prod_{k=1}^{K} q_k + \sum_{i<j} \kappa_{ij} + \sum_{i<j<k} \kappa_{ijk} + \sum_{i<j<k<\ell} \kappa_{ijkl} + \cdots + \kappa_{1,2,\dots,K} \quad (1)$$

The dots represent cumulants of order four and higher. equation 1 can be written compactly as

$$\mathbb{E}\left[\prod_{k=1}^{K} F_k\right] = \sum_{I \subseteq [K]} \kappa_I,$$

where $\kappa_I$ is the joint cumulant on the index set $I$ (with $\kappa_{\{i\}} = q_i$ and $\kappa_\varnothing = 1$). □

## E  MORE RESULTS

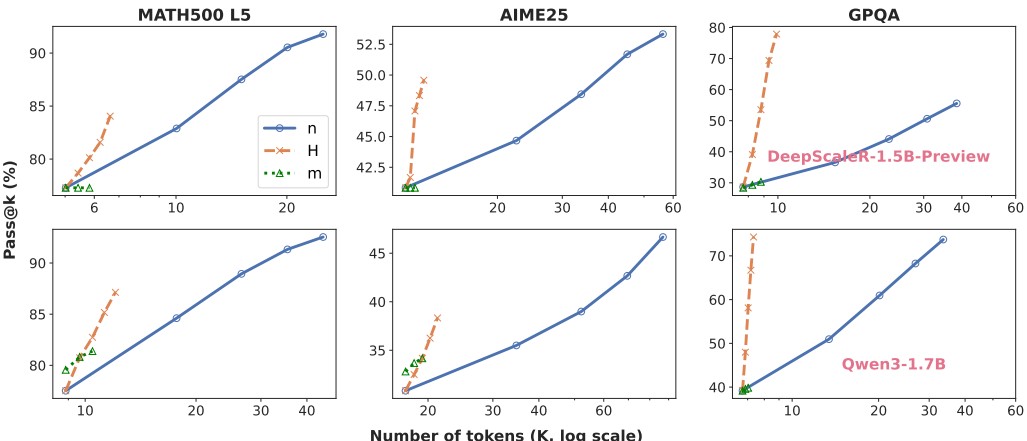

Figure E.1: Pass@$k$ performance versus total token budget for three sampling schemes In each subplot, we compare: **m** (green dotted)–sampling only the final solution; **n** (blue solid)–sampling full reasoning trajectories; **H** (orange dashed)–fractured sampling across all intermediate steps. Rows correspond to DeepScaleR-1.5B-Preview and Qwen3-1.7B models models. Fractured sampling (H) consistently yields higher pass@$k$ at a given token budget.

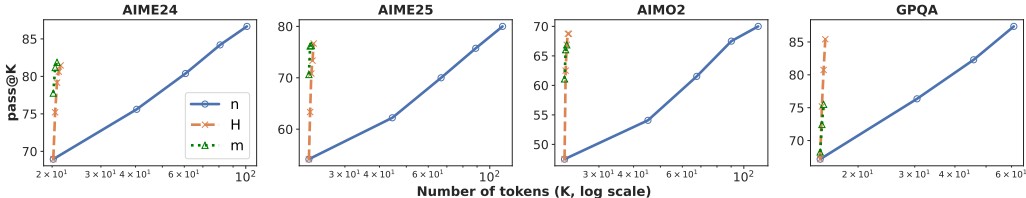

Figure E.2: Pass@$k$ performance versus total token budget for three sampling schemes on GPT-OSS-20B. In each subplot, we compare: **m** (green dotted)–sampling only the final solution; **n** (blue solid)–sampling full reasoning trajectories; **H** (orange dashed)–fractured sampling across all intermediate steps. Fractured sampling (H) consistently yields higher pass@$k$ at a given token budget.

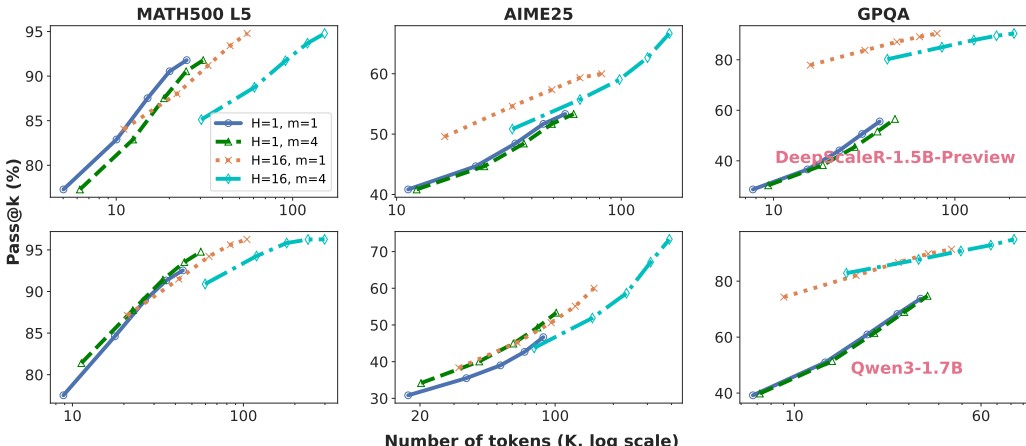

Figure E.3: Pass@$k$ performance versus total token budget for four sampling schemes on five benchmarks. In each subplot, we compare: **H=1, m=1**–only sampling full reasoning trajectory; **H=1, m=4**–sampling both full reasoning trajectories and the final solution; **H=16, m=1**–both full reasoning trajectory sampling and fractured sampling across all intermediate steps; **H=16, m=4**–sampling all three dimensions. $n$ is in [1, 2, 4, 8, 16] for the five points (from left to right) on each line. Rows correspond to DeepScaleR-1.5B-Preview and Qwen3-1.7B models. MATH500 L5 is saturated here, resulting in a less efficient gain from dimensions $H$ and $m$.

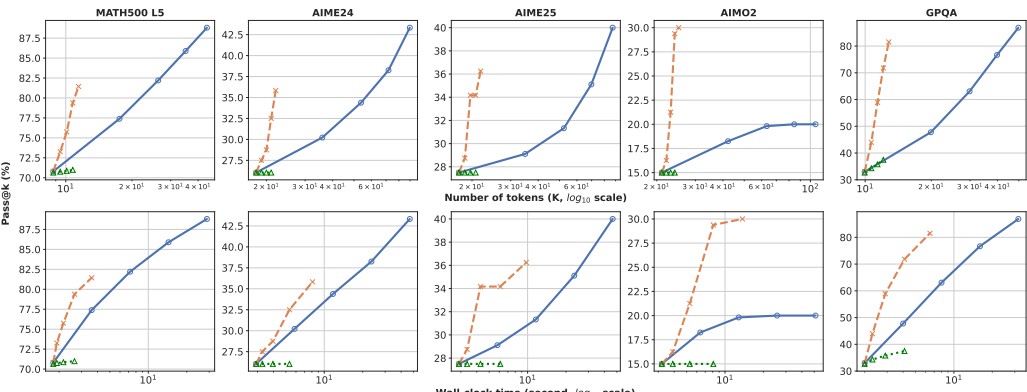

Figure E.4: Pass@$k$ performance versus token budget or wall-clock time. We compare: **m**–sampling only the final solution; **n**–sampling full reasoning trajectories; **H**–fractured sampling across all intermediate steps. DS-R1-Qwen-1.5B is utilized here. Fractured sampling consistently yields higher pass@$k$ at a given token or time budget.

