# OpenReview forum: "Fractured Chain-of-Thought Reasoning"
_ICLR.cc/2026/Conference — Submitted to ICLR 2026_

### Official Review · Reviewer_BfLE · 2025-10-26

**Soundness:** 2
**Presentation:** 2
**Contribution:** 3
**Rating:** 2
**Confidence:** 4

**Summary:**

This paper challenges the common assumption that complete, long chain-of-thought (CoT) traces are necessary for accurate reasoning. The authors show that incomplete or truncated CoT trajectories can still yield highly accurate results—a concept they term Fractured Sampling. They introduce Depth of Reasoning (H) as a new sampling axis, complementing existing dimensions of Trajectory Sampling (n) and Solution Sampling (m). Empirical validation is conducted on five challenging reasoning benchmarks (MATH500 Lv5, AIME24, AIME25, AIMO2, GPQA) across multiple model scales (e.g., DeepSeek-R1, Qwen, etc.), demonstrating consistent improvements under equal token budgets.

**Strengths:**

- The paper identifies an under-explored dimension of inference-time sampling in CoT-based LLM reasoning—not only how many reasoning chains or final answers to generate, but where in the reasoning trace sampling should occur.
- The empirical evaluation spans several mathematical and scientific reasoning benchmarks, using models of various scales, showing consistent and interpretable trends.
- The method operates purely at inference time without requiring retraining, making it practical for latency- or resource-constrained settings.
- The concept of fractured sampling (sampling across reasoning depth) is novel and provides new insights into inference-time scaling laws.

**Weaknesses:**

- Line 128: Pass@k is introduced as one of the sampling schemes. However, Pass@k is an evaluation metric that requires access to ground-truth answers, not a sampling method. Unless I am misunderstanding, please clarify this distinction in the text.

- Line 152: Mathematical notations are overloaded. The use of ε with multiple subscripts (sometimes εᵢ, sometimes εᵢⱼ) and later with a superscript (Line 169) reduces readability. Please standardize these notations to avoid confusion.
- The token-budget modeling (B(n, m, H) = n·C_thinking + n·m·H·C_solution) assumes uniform cost per reasoning step or solution. In practice, later prefixes may incur different costs, and generation speeds may vary. The resulting cost–performance curves could differ in real-world compute regimes, particularly under GPU batching, caching, or branching overheads.
- The finding that truncated CoT (i.e., stopping reasoning early and directly generating the answer) often matches full CoT accuracy (Figure 2) is intriguing but raises an important question: Why does additional reasoning not always improve accuracy? The paper includes correlation analysis but could benefit from deeper investigation into failure modes—for instance, at what prefix length correct answers typically emerge.
- Some benchmarks (e.g., AIME with only 30 samples) are relatively small, which may reduce statistical robustness. In addition, Pass@k (measuring whether at least one of k samples is correct) does not penalize low diversity or duplicate answers.
- While the paper focuses on accuracy metrics such as Pass@k, the quality, coherence, or interpretability of truncated reasoning traces is not analyzed. For tasks requiring human-auditable reasoning, shorter CoTs may yield less informative or less interpretable explanations.
- Line 309: The text states that the entire reasoning trace is first generated and then divided into H equal-sized segments (based on token count, e.g., H = 16). This raises an important question: during inference, how should one choose H? If the full trace must already be generated before segmentation, how does this method actually save tokens or computation?

**Questions:**

- Figures 4 & 5 (Log Scale): The x-axes in Figures 4 and 5 are plotted in log scale. The meaning of the “60” mark on the x-axis is unclear.
For example, if n = 16 and the maximum token limit is 32 768, the total token budget would be 524 288. The natural logarithm of this value is approximately 13, not 60. Please clarify how the horizontal axis is normalized or scaled (e.g., whether it represents log₁₀ of tokens, or scaled thousands of tokens).
- Please define all model abbreviations explicitly. For example, clarify that “DS-R1-1.5B” refers to DeepSeek-R1-Distill-Qwen1.5B, and explain what “DS-R1” alone represents.
- A more explicit description or pseudocode of the procedure would improve clarity.

---

> ### Author Response · Authors · 2025-11-21
> **Response to Reviewer BfLE (part 1)**
>
> Dear Reviewer BfLE,
>
> Thank you very much for your time, effort, and thorough review. We address your concerns below:
>
> $~$
>
> ---
>
> > W1. “Line 128: Pass@k is introduced as a sampling scheme, but Pass@k is an evaluation metric.”
>
> Thank you for pointing this out. We agree this phrasing was imprecise.
> - We do not use Pass@k as a sampling method.
> - In the affected sentence, “Pass@k” referred to sampling multiple candidate solutions and then evaluating via Pass@k.
>
> We have rephrased the paragraph title (L104) from "Baseline sampling schemes" to "Baseline inference techniques" in the updated version.
>
> $~$
>
> > W2. "Line 152: Notation is overloaded (εᵢ, εᵢⱼ, ε with superscripts). Standardize."
>
> We appreciate the feedback. The goal was to distinguish the randomness along different dimensions. I.e., $i$, $j$ and $t$ are for the $n$, $m$ and $H$ dimension, respectively. We have standardized the notation, and only utilize subscript in the updated version (L152, L190).
>
> $~$
>
> > W3. "Token-budget model assumes uniform cost per reasoning step or solution. In practice, later prefixes may incur different costs, and generation speeds may vary; real cost may vary."
>
> This is indeed a practical concern.
>
> As mentioned that "the resulting cost–performance curves could differ in real-world compute regimes, particularly under GPU batching, caching, or branching overheads". With different GPU settings (GPU type, number of GPUs, batch size, and so on), the scaling law might be varied. Using the number of tokens as a budget estimation seems more reliable and robust.
>
> There might be a misunderstanding that we don't assume a uniform cost per reasoning step. Instead, we assume a uniform cost per solution given different length of prefixes. The reason is: the solution tokens only account for <7% of total tokens. To justify our assumption, we show the wall-clock time required by the solutions at different depth. The experimental settings here are:
> - Model: DS-R1-Qwen-7B
> - Task: MATH500 L5 and AIME24
> - GPU: one A100-80GB
> - vLLM: default batch size and caching setting.
> - We average the wall-clock time per solution across number of samples with 3 random runs.
>
> As shown in the following table, the wall-clock time is very close for solutions generated at different depths. We argue that the different length of prefixes doesn't have a significant influence to the sampling of short solutions.
>
> | Depth $H$ | 1 | 2 | 4 | 8 | 16 |
> | --- | --- | --- | --- | --- | --- |
> | Time required by each solution (second) | 2.54 | 2.52 | 2.49 | 2.50 | 2.49 |
>
> $~$
>
> In addition, we add another scaling law figure w.r.t. latency, i.e. Figure E.4 in appendix (L888-L905) in the updated version. In Figure E.4, we use 8 A100-80GB, and the default batch size and caching setting from vLLM. **The observation from latency is consistent to the one from the number of tokens: $H$ dimension shows the most efficient gain.**
>
> $~$
>
> > W4. "Why does additional reasoning not always improve accuracy? Deeper investigation needed. For instance, at what prefix length correct answers typically emerge."
>
> Great question. Our correlation analysis partially addresses this, but we agree deeper explanation helps.
>
> The key empirical finding is:
> - Correct answers typically emerge earlier than the point where reasoning becomes long and verbose.
> - Later reasoning steps often introduce overthinking errors, hallucinated algebraic manipulations, or unnecessary branching—an effect also observed in the Qwen3 paper.
>
> To show the prefix length where a correct answer typically emerge, we (1) select the samplings with correct answers; (2) record the number of reasoning tokens along the $H$ dimension ($H=16$) required for the first correct answer, and divide by the total number of tokens of full trace. We use DS-R1-Qwen-1.5B here.
>
> The following table shows:
> - Only <75% reasoning tokens are needed across various tasks;
> - The correct answer for the easier tasks (MATH500 and GPQA) tend to happen more early, which is reasonable because such tasks should require less reasoning.
>
> | MATH500 L5 | AIME24 | AIME25 | AIMO2 | GPQA |
> | --- | --- | --- | --- | --- |
> | 58.7% | 73.2% | 75.2% | 73.5& | 64.3% |
>
> $~$
>
> > W5. "Some benchmarks (AIME) are small."
>
> We agree with this point. It is also the main reason for us to include 5 benchmarks (MATH500, AIME24, AIME25, AIMO2 and GPQA) to verify the effectiveness of our method. We tried to use other benchmarks, like GSM8K and AMC. They are too easy for the reasoning models.
>
> Here we further include a code benchmark, LiveCodeBench, with 175 prompts. Coding task doesn't support majority voting, so we show the Best-of-N performance on DeepSeek-R1-Distill-Qwen1.5B. We can observe: The $H$ dimension shows the most efficient gain.
>
> | H | m | Accuracy |
> | --- | --- | --- |
> | 1 | 1 | 19.3 |
> | 16 | 1 | 22.1 |
> | -4 | 1 | **23.2** |
> | 1 | 4 | 19.3 |
> | 16 | 4 | **23.2** |
> | -4 | 4 | **23.2** |
>
> ---
>
> $~$
>
> Please move to part 2.

---

> ### Author Response · Authors · 2025-11-21
> **Response to Reviewer BfLE (part 2)**
>
> > W5. "Pass@k may not penalize low diversity."
>
> We agree with this point. Diverse predicitons benefit Pass@k. This is also the main claim from our paper. I.e. the $H$ dimension encourages the sampling diversity.
>
> However, we also report majority voting and Best-of-N in Table 1, which are sensitive to diversity and consistency. We show that Fractured Sampling works well for both kinds of metrics.
>
> $~$
>
> > W6. "Quality/interpretability of truncated reasoning not analyzed."
>
> This is an important point. Our focus is on accuracy–cost trade-offs, but we agree interpretability is a separate axis.
>
> We note:
> - Fractured Sampling does not reduce the quality of full reasoning traces; it only samples solutions given shorter prefixes.
> - For interpretability-sensitive tasks, users can choose to display the full final reasoning even when sampling intermediate solutions.
>
> $~$
>
> > W7. "During inference, how should one choose H?"
>
> **Short Answer: Ensuring each reasoning step to have a moderate number of tokens (2K tokens) is a practical setting across various LLMs and tasks.**
>
> As shown in Figure 2(b), the solution tokens only account for <7% of total tokens across various models and tasks. Along the $H$ dimension, we only sample solutions and don't introduce new thinking tokens. The most tokens are consumed on the $n$ dimension (similar to self-consistency). Therefore, during inference, a trial-and-error on $H$ is very efficient. We also show a practical guidance in Section 4.4, i.e., ensuring each reasoning step to have a moderate number of tokens, like 2K. I.e., we sample a new solution afer each 2K reasoning tokens. All evaluated LLMs and tasks show good results for this setting.
>
> $~$
>
> >  W7. "If the full trace must be generated before segmentation, how does Fractured Sampling save tokens?"
>
> When showing the scaling law in Figure 4 and 5, we first generate the full trace, and then split. In this way, we can strictly control the number of tokens for each reasoning step, making the scaling law more reliable and robust.
>
> Although we need to generate the full trace first, it still saves tokens. In Figure 4, the setting for the $H$ line is "$n=1, H \in[1,2,4,8,16]$". I.e. we only sample one full trace. The setting for the line $n$ is "$n\in[1,2,4,8,16], H=1$". I.e. we sample multiple full traces. Since the $H$ line only requires one full trace, it consumes much less tokens than the $n$ line.
>
> In Section 4.4, we also show an early stopping method where we don't need to sample the full trace. We only need to sample the solution after each 2K reasoning tokens, and stop the reasoning if the prediciton converges. This method saves ~20% tokens while maintaining the accuracy.
>
> $~$
>
> > Q1. "Figures 4 & 5: log-scale x-axis unclear; why does it show 60 instead of ~13?"
>
> We use the default log scale from the library Matplotlib, i.e. $log_{10}$. We have mentioned this setting in the updated caption.
>
> $~$
>
> > Q2. "Define model abbreviations (e.g., DS-R1-1.5B, DS-R1)."
>
> DS-R1 refers to DeepSeek-R1 with 685B parameters. Thank you for this suggestion. We have added the model definition in L347-L349.
>
> ---
>
> $~$
>
> Please move to part 3.

---

> ### Author Response · Authors · 2025-11-21
> **Response to Reviewer BfLE (part 3)**
>
> > Q3. "Provide pseudocode or more explicit description of the procedure."
>
> This is a great suggestion. For a better demonstration of our method, we:
> - add Algorithm 1 (L162-L180) in the updated version,
> - add a pseudocode in python as following;
> - add a short  GIF in the supplementary materials. You only need to download the supplementary materials, open with VSCode, and view README.
>
> Note: The pseudocode here is for the demonstration purpose. The real implementation (code is in the supplementary materials) will do the sampling in a much more efficienct way:
>
> ```python
>
> def UNIFIED_SAMPLING(prompt, n, m, H, selector):
>     """
>     prompt   : input question
>     n        : number of full trajectories
>     H        : number of depth segments within each trajectory
>     m        : number of solutions sampled per segment
>     selector : final-answer selector (PRM, majority vote, etc.)
>     """
>
>     all_candidates = []   # (answer, depth, trajectory_id)
>
>     # === 1. Generate n full CoT trajectories ===
>     full_trajs = []
>     for traj_id in range(n):
>         full_cot = MODEL.generate_full_cot(prompt)  # long reasoning trace
>         full_trajs.append(full_cot)
>
>     # === 2. Partition each trajectory into H equal-length segments ===
>     # (based on token length, ensures comparability across trajectories)
>     for traj_id, cot in enumerate(full_trajs):
>         tokens = tokenize(cot)
>         T = len(tokens)
>         segment_size = max(1, T // H)
>
>         for h in range(1, H+1):
>             # prefix ending at segment h
>             end = min(T, h * segment_size)
>             prefix_tokens = tokens[:end]
>             prefix = detokenize(prefix_tokens)
>
>             # === 3. At this depth, sample m solutions ===
>             for j in range(m):
>                 solution = MODEL.generate_solution(prefix, prompt)
>
>                 # === 4. Extract the final answer from the solution ===
>                 answer = extract_answer(solution)
>
>                 all_candidates.append({
>                     "answer": answer,
>                     "depth": h,
>                     "trajectory": traj_id
>                 })
>
>     # === 5. Apply selection mechanism ===
>     final_answer = selector(all_candidates)
>
>     return final_answer, all_candidates
> ```
> ---
>
> $~$
>
> Thank you again for your thoughtful question and suggestion. Should you have any further questions, we are happy to assist.

---

### Official Review · Reviewer_CQMk · 2025-10-27

**Soundness:** 2
**Presentation:** 3
**Contribution:** 2
**Rating:** 4
**Confidence:** 3

**Summary:**

This paper introduces Fractured Sampling, an inference-time strategy to improve LLM reasoning efficiency and accuracy. The authors first observe that generating a truncated CoT reasoning trace can achieve accuracy comparable to a full CoT, while using far fewer tokens. Building on this insight, Fractured Sampling is proposed to interpolate between solution-only decoding and full CoT by exploring three axes: trajectory count, solution count per path, and reasoning depth. By controlling these dimensions, the approach generates a diverse set of candidate answers at various intermediate reasoning stages, leveraging the model’s internal thought process. Extensive experiments on five challenging reasoning benchmarks across several model scales demonstrate that Fractured Sampling consistently yields a better accuracy-versus-cost trade-off than standard decoding strategies. Notably, under the same token budget, this method achieves higher success rates than both solution-only sampling and full CoT sampling, effectively shifting the inference scaling curve upward. The paper provides theoretical analysis explaining these gains: sampling at intermediate reasoning steps exposes diverse error modes that are less correlated with each other, thereby increasing the probability of finding at least one correct answer. The authors also analyze how to optimally allocate a fixed token budget across the three axes, finding that devoting more budget to reasoning-depth branching typically yields the greatest benefit.

**Strengths:**

Novel Method: The idea of sampling intermediate reasoning steps is novel and expands on prior inference-time techniques by introducing a new axis of diversity. Unlike conventional decoding which samples only complete solutions or final answers, Fractured Sampling explicitly fractures the reasoning process, aggregating partial reasoning outcomes. This unified framework can be seen as a fine-grained TOT approach where each branch corresponds to a partial reasoning prefix.

Theoretical Analysis: The paper provides a solid theoretical explanation for why Fractured Sampling works. It derives a lower-bound on the success probability by considering the diversity of failure events across the K sampled branches. The key insight is that if errors at different reasoning depths are not perfectly correlated, then sampling across those depths markedly reduces the chance that all branches fail. The authors back this up with empirical correlation matrices showing that many pairs of depth positions have low or negative error correlation. This analysis is convincing and aligns with the experimental finding that tasks with more diverse error patterns across reasoning steps gain the most from the fractured approach.

Practical Insights and Guidelines: Beyond raw results, the paper offers valuable insights for practitioners. It identifies that most of the token budget in CoT reasoning is spent on the thinking steps, not on final answers. This justifies the focus on optimizing reasoning depth. The scaling-law analysis reveals a roughly log-linear improvement in success as more compute is spent, and that Fractured Sampling yields the steepest slope among the methods. The authors distill a clear guideline: if you have a fixed token budget, invest in branching along the reasoning depth first, before adding more solution samples, as the former gives higher payoff in accuracy. Such guidance is extremely useful and demonstrates the authors’ deep understanding of the method’s behavior.

**Weaknesses:**

Complexity and Implementation Overhead: Fractured Sampling introduces additional complexity to the inference process. It requires controlling the generation to stop at multiple intermediate points and branching out multiple final answers from each, which in practice means many forward passes or a more complex decoding procedure. This could be cumbersome to implement, especially in black-box API settings where one cannot easily intervene mid-generation. The authors note that their approach assumes access to the model’s internal sampling process, which may not be feasible for all users. Even with access, the total number of model evaluations is n * H * m for full three-dimensional sampling. If not managed carefully, this could negate the token savings or increase latency. The paper does not explicitly discuss the wall-clock latency implications; a brief analysis of real-time speed vs. accuracy would further strengthen the practicality argument.

Tuning of Hyperparameters: Using Fractured Sampling effectively might require tuning the axes (n, m, H) for different tasks or models. The authors provide general guidance that allocating budget to H is most efficient, and they use fixed values like H=16, m=4 in experiments. However, an open question is how to choose the truncation depth H optimally for a new task or how sensitive performance is to this choice. In one experiment, they found that including too many early-step solutions introduced noise for the reward model selector, which they solved by heuristically discarding the first 75% of steps. This suggests some trial-and-error in finding the right balance. A more systematic approach or discussion on setting these hyperparameters would be beneficial for reproducibility.

Limited Discussion of Potential Downsides: The paper could discuss more the potential downsides or failure cases of Fractured Sampling. For example, one can imagine that if a model tends to make a particular systematic error early in reasoning, branching on that step might propagate the same error to many final answers. The authors do not report significant cases where more depth branching hurt performance. It would be insightful to know if there are tasks where adding intermediate branches yields diminishing returns or even confusion. Additionally, generating many candidate answers could complicate answer selection. The paper uses a strong process reward model and majority voting to pick final answers, which may not be available in all settings. These concerns do not appear to outweigh the benefits in the tested scenarios, but acknowledging them would provide a more balanced evaluation.

**Questions:**

Generality to Other Domains: Have you tested or do you anticipate Fractured Sampling to be effective on non-mathematical reasoning tasks? The current benchmarks are math and science-heavy. If a task doesn’t naturally involve multi-step reasoning or if the model tends not to produce a long CoT, how would the method behave? Any insight into applying FS in those contexts would be useful.

Dynamic Truncation: How was the truncation depth H decided for your experiments? Could H be set dynamically per query? You implement an early-stopping heuristic based on answer convergence. Could this idea be extended to adapt the truncation depth during generation instead of using a fixed number of steps for all queries?

Model Guidance: When generating intermediate solutions at depth t, did you simply prompt the model to output an answer directly after the truncated reasoning, or did you use any special prompting? Clarifying the implementation would help: e.g., do you run the model separately for each (prefix, final-answer) pair, or generate the full reasoning once and split it? I wonder if there is a way to get the model to “decide” it has enough information at depth t to answer, perhaps using a trained value function or uncertainty measure. This could potentially reduce generating very uncertain early answers. Any thoughts on guiding the model’s internal decision of when to answer would be interesting.

Selection Mechanism Robustness: In your evaluation, you rely on a process reward model to pick the best answer and also use majority voting. Did you notice any cases where these selection mechanisms failed or were biased by the presence of many partial solutions? The result where using all H=16 steps slightly hurt PRM performance suggests that lower-quality early answers can introduce noise. Your solution was to filter out early steps. Could another solution be to weight answers by depth or confidence? More generally, how critical is the choice of PRM and could a weaker selection model diminish the gains of Fractured Sampling?

Integration in APIs: You mention that Fractured Sampling assumes low-level control over the model’s generation process. For commercial LLM APIs that don’t allow iterative prompting at each reasoning step, do you have suggestions on how to approximate Fractured Sampling? Do you think the benefits of FS could be attained in such scenarios?

---

> ### Author Response · Authors · 2025-11-21
> **Response to Reviewer CQMk (part 1)**
>
> Dear Reviewer CQMk,
>
> Thank you very much for your time, effort, and thorough review. We address your concerns below:
>
> $~$
>
> ---
>
> > W1. Complexity & Implementation Overhead (1A): "Fractured Sampling requires controlling the generation to stop at multiple intermediate points and branching out multiple final answers from each, which in practice means many forward passes or a more complex decoding procedure."
>
> Thank you for highlighting this practical concern. We would like to clarify that Fractured Sampling introduces only minimal additional implementation complexity compared to standard self-consistency:
>
> - Self-consistency ($n$-axis only): The same prompt is repeated $n$ times, and the LLM samples the full trajectory (reasoning + solution) for each prompt with different seeds. It requires one forward pass, if all prompts can be fed into one batch.
> - Fractured Sampling ($H$-axis only): We generate one reasoning trajectory, split it into H segments with equal length (no custom stop signal needed), and then generate solutions from each prefix. This involves one forward pass for reasoning and one lightweight forward pass for solutions. The latter is inexpensive because, as shown in Figure 2(b), solution tokens only account for <7% of total tokens.
>
> In other words, Fractured Sampling requires only a single additional short pass, and does not increase the cost of the reasoning portion. Given that Fractured Sampling provides substantially better test-time scaling performance than self-consistency alone (Figure 4), we believe this small implementation overhead is well justified.
>
> $~$
>
> > W1. Complexity & Implementation Overhead (1B): "Hard to implement in black-box APIs that don't allow mid-generation control and don't show visible thinking/reasoning."
>
> We agree that Fractured Sampling assumes access to intermediate reasoning states. Fortunately, this capability is widely available in open-source models and infrastructures, such as vLLM, SGLang, LMDeploy, and HuggingFace generation hooks. We view the access to intermediate reasoning states as a minor prerequisite for two reasons:
>
> - Limiting research design to what closed-source APIs expose would restrict the contributions we can make to the open-source community.
> - Many prior works on test-time scaling [1,2,3], efficient reasoning sampling [4], and long-CoT compression [5, 6] rely on exactly the same assumption. Our requirement is therefore aligned with standard practice in this research area.
>
> $~$
>
> [1] s1: Simple test-time scaling, Niklas Muennighoff, Zitong Yang, Weijia Shi, ...
>
> [2] SCALING LLM TEST-TIME COMPUTE OPTIMALLY CAN BE MORE EFFECTIVE THAN SCALING PARAMETERS FOR REASONING, Charlie Snell, Jaehoon Lee, Kelvin Xu, Aviral Kumar
>
> [3] AlphaMath Almost Zero: Process Supervision without Process, Guoxin Chen, Minpeng Liao, Chengxi Li, Kai Fan
>
> [4] Beyond the Last Answer: Your Reasoning Trace Uncovers More than You Think, Hasan Abed Al Kader Hammoud, Hani Itani, Bernard Ghanem
>
> [5] TokenSkip: Controllable Chain-of-Thought Compression in LLMs, Heming Xia, Chak Tou Leong, Wenjie Wang, Yongqi Li, Wenjie Li
>
> [6] CoT-Valve: Length-Compressible Chain-of-Thought Tuning, Xinyin Ma, Guangnian Wan, Runpeng Yu, Gongfan Fang, Xinchao Wang
>
> $~$
>
> > W1. Complexity & Implementation Overhead (1C): "The total number of model evaluations is $n * H * m$ for full three-dimensional sampling. If not managed carefully, this could negate the token savings or increase latency."
>
> Among all three dimensions, only the $n$ dimension (like self-consistency) requires the sampling of a full reasoning trajectory, while $H$ and $m$ only sample the solution part. As shown in Figure 2(b), the solution tokens only account for <7% of total tokens. We believe the token consumption prediction is similar to self-consistency. Especially with our best setting ($H=-4$) from Table 1, the increased number of tokens is well under management.
>
> $~$
>
> > W1. Complexity & Implementation Overhead (1D): "Wall-clock latency was not discussed."
>
> This is indeed a practical concern. We mainly draw the scaling law based on the token budget, because it is more general and robust. For wall-clock latency, it might vary given different GPU types, number of GPUs, batch sizes and caching setting.
>
> However, we agree that adding the wall-clock latency helps readers to deeply undertand our work. **We add a new figure, Figure E.4 (L888-L905),  to the updated version. We can notice that the finding from token budget still holds for the time budget.** The $H$ dimension shows the best accuracy per second.
>
> Experimental setting: We use 8 A100-80GB, and the default batch size and caching setting from vLLM.
>
> ---
>
> $~$
>
> Please move to part 2.

---

> ### Author Response · Authors · 2025-11-21
> **Response to Reviewer CQMk (part 2)**
>
> > W2. Tuning of Hyperparameters: "Choosing n, m, H requires trial-and-error."
>
> This is indeed a very practical concern.
>
> Our method shares the same trial-and-error as self-consistency. For self-consistency, the best setting of $n$ varies for different tasks. For example, a smaller $n$ is enough for a simple task, while difficult task requires a much higher $n$.
>
> Fortunately, the tuning of our $H$ dimension is very efficient. We sample multiple solutions along the $H$ dimension. As shown in Figure 2(b), the solution tokens account for <7% of total tokens. The tuning of $H$ is much efficient than the $n$ dimension, because the $n$ dimension requires the sampling of a full reasoning trace.
>
> In Section 4.4, we also empirically show a well-performing setting: sampling the solution after each 2K reasoning tokens. In this setting, we don't need to set a explicit $H$ for different tasks. The LLM dynamically allocate the reasoning tokens:
> - If a task is very simple, the first two predictions are likely to be the same. And we terminate the sampling. Therefore, we use fewer tokens for easy task.
> - If a task is very difficult, the first few predictions are noisy and less likely identical. The prediction only converges at the end of reasoning. Therefore, we use more tokens for difficult task.
>
> $~$
>
> > W3 and Q4. Limited Discussion of Potential Downsides. "how critical is the choice of PRM and could a weaker selection model diminish the gains of Fractured Sampling?"
>
> We totally agree that the discussion of potential limitation could provide more insight to the reader, and help the follow-up work. In our paper, we honestly reveal the challenges we met:
> - In section 4.3, we reveal that the early prediction is noisier;
> - In Appendix C (Limitations), we reveal that our framework requires the accesss to the reasoning path.
>
> Regarding to the mentioned limitation or downsides, we address them as folowing:
>
> 1. **"Branching on a early step with a systematic error might propagate the same error to many final answers."**
>
> A common assumption for the better performance from reasoning model than non-reasoning model is: the long CoT from reasoning model contains more reflection and self-correction steps. It means that the error happening at a early step could be potentially fixed after more reasoning. Our Fractured Sampling doesn't modify the long CoT. Instead, we sample multple solutions given different length of prefixes. I.e. the later solution is sampled with more reasoning that could fix the early error.
>
> Figure 3 also supports our claim: Many entries are light or even pink (negative), signalling that failures at two distinct depths tend not to happen simultaneously.
>
> 2. **"The authors do not report significant cases where more depth branching hurt performance. It would be insightful to know if there are tasks where adding intermediate branches yields diminishing returns or even confusion."**
>
> We reported it in Table 1, where $H=16$ achieves slightly worse accuracy than $H=1$ for majority voting (65.3 vs 66.7). Because majority voting puts the same weight to all predictions. And the noisier predicitons from the early steps confuse the selection.
>
> 3. **"Additionally, generating many candidate answers could complicate answer selection."**
>
> Compared to self-consistency (i.e. majority voting) and Best-of-N, our method doesn't introduce overhead to the answer selection. As shown in Table 1, both majority voting and Best-of-N works for our framework.
>
> 4. **"The paper uses a strong process reward model and majority voting to pick final answers, which may not be available in all settings."**
>
> This is a very practical concern, and a common challenge for test-time scaling methods. And it is also the reason for us to include both majority voting and Best-of-N to justify the effectiveness of our framework.
> - Majority voting: For tasks with objectively checkable and discrete answer, like MATH500, GPQA, AIME and so on.
> - Best-of-N: This is a more general answer selection method, if the reward model performs well. We can use it for both verifiable and non-verifiable tasks.
>
> Here we further include a weaker PRM, Qwen2.5-Math-PRM-7B. As shown in the following table, a weaker PRM still works for the answer selection.
>
> Lastly, we are also grateful for your acknowledgement that "These concerns do not appear to outweigh the benefits in the tested scenarios".
>
> | H | m | MATH500 L5 | AIME24 | AIME25 | AIMO2 | GPQA | Avg. |
> | --- | --- | --- | --- | --- | --- | --- | --- |
> | 1 | 1 | 90.3 | 62.5 | 53.3 | 40.0 | 53.5 | 59.9 |
> | -4 | 1 | 93.3 | 70.0 | **60.0** | 50.0 | 54.0 | 65.5 |
> | 1 | 4 | 90.3 | 70.0 | 53.3 | 40.0 | 55.2 | 61.8 |
> | -4 | 4 | **94.2** | **73.3** | **60.0** | **60.0** | **55.9** | **68.7** |
>
> ---
>
> $~$
>
> Please move to part 3.

---

> ### Author Response · Authors · 2025-11-21
> **Response to Reviewer CQMk (part 3)**
>
> > Q3. Model Guidance (moved forward for better rebuttal): "The implementation detail of Fractured Sampling. Do you run the model separately for each (prefix, final-answer) pair, or generate the full reasoning once and split it?"
>
> Sorry for this confusion. In the supplementary materials, we added a short GIF for your easy understanding. You only need to download the the supplementary materials, open the codebase with VSCode, and then view README. (The frac_cot.gif locates in figs folder.)
>
> Here we take DeepSeek-R1 as an example for explanation.
>
> Given a prompt $p$, the LLM's output is in this format: "\<think\> $h_1, ..., h_t, ..., h_H$ \</think\> $z$". At depth $t$, we simply append a "\</think\>". I.e. the input to LLM becomes "$p$ \<think\> $h_1, ..., h_t$ \</think\>". A new solution $z_t$ is sampled with this truncated CoT.
>
> As you can see, we don't need any specific prompting. Only a simple cut with a appending of "\</think\>" is enough. Such a design is very simple for implementation, introducing minimal overhead.
>
> Specifically, Fractured Sampling consists of two steps:
> 1. First generate the full reasoning, i.e. "\<think\> $h_1, ..., h_t, ..., h_H$ \</think\>"
> 2. Sample solutions with different prefixes. This step only requires one forward pass, because we can input them in a batch as:
>
> ["$p$ \<think\> $h_1$ \</think\>", ..., "$p$ \<think\> $h_1, ..., h_t$ \</think\>", ...,  "$p$ \<think\> $h_1, ..., h_t, ..., h_H$ \</think\>"].
>
> Notablly, for this list of inputs, we don't need to recompute the prefixes. We can enable prefix caching for step 1, and the KV caching are shared to step 2.
>
> $~$
>
> > Q1. Generality to Other Domains: "Generalization of Fractured Sampling to tasks with less reasoning."
>
> This is a very great suggestion.
>
> **Short Answer: Fractured Sampling can seamlessly work for any LLM that has above-mentioned DeepSeek-R1's output format. If the LLM is a short-CoT model without "\<think\> \</think\>", we can simply append "The final answer is" instead of "\</think\>" after the truncated CoT. For tasks with less reasoning, we can set a smaller $H$.**
>
> If a task doesn't require intensive reasoning, the CoT becomes much shorter. For such a case, setting a large $H$ is less reasonable. For example, if the number of thinking tokens for a task is 512 and we set $H=16$, the later truncated CoT only has 512/16=32 more reasoning tokens, less likely to introduce new reasoning. Setting a small $H$ makes more sense.
>
> Here we apply Fractured Sampling to a creative writing task, CreativeWritingV3, with Llama-3.1-8B-RLMT [7]. We set $H=2$ and $H=4$, since the number of thinking tokens are commonly <1K for CreativeWritingV3. Skywork-V2 is used for answer selection for $n>1$ or $H>1$. GPT-4.1 is used as a judge to score the writing (after selection) between 0-100. Please refer to [7] for more details.
>
> **As shown in the following table, Fractured Sampling still works for task requiring less reasoning. Setting a larger $H$ offers a minimal gain, because the CoT is much shorter than math or science tasks.**
>
> According to our experience from math, science and writing tasks, setting $H$ to ensure each reasoning step to have a proper length (>512 for reasoning-less task, and >2048 for reasoning-intensive task) generally has a better performance. It also offers a flexible budget control option to the user.
>
> | n | H | score |
> | --- | --- | --- |
> | 1 | 1 | 80.9 |
> | 2 | 1 | 82.3 |
> | 1 | 2 | 83.0 |
> | 1 | 4 | **83.1** |
>
> [7] LANGUAGE MODELS THAT THINK, CHAT BETTER, Adithya Bhaskar, Xi Ye, Danqi Che
>
> $~$
>
> > Q2. Dynamic Truncation
>
> 1. **"How was the truncation depth H decided for your experiments?"**
>
> In Figure 4, we set both $n$ and $H$ in [1, 2, 4, 8, 16] to show the scaling behavior. Due to computation limit, we couldn't further explore a larger $n$ and $H$. From Figure 4, we notice that the performance from $H=16$ is still not plateaued. Thus for Figure 5 and Table 1, we set $H=16$ by default.
>
> 2. **"Could H be set dynamically per query? You implement an early-stopping heuristic based on answer convergence. Could this idea be extended to adapt the truncation depth during generation instead of using a fixed number of steps for all queries?"**
>
> Yes, Section 4.4 already introduces a dynamic early-stopping mechanism, which is precisely a per-query adaptive $H$. For example:
> - If a question is simple, and a premature thinking could already offer a valid reasoning. The predictions from the first two steps are mostly identical. For such a case, $H=2$.
> - If a question is difficult, and requires a very intensive and long reasoning. The early predictions are more likely noisy and different. The identical answer happens much later. For such a case, $H$ is larger.
>
> This is a very simple and easy-to-implement method for dynamic token allocation per query.
>
> ---
>
> $~$
>
> Please move to part 4.

---

> ### Author Response · Authors · 2025-11-21
> **Response to Reviewer CQMk (part 4)**
>
> > Q3. Model Guidance: "Any thought on guiding the model’s internal decision of when to answer would be interesting."
>
> This is a great question! Determining when to stop over-thinking and make an early answer is a fundamental solution to efficient long-CoT sampling. Here we offer two potential designs:
> - **Aggregated token-level entropy**: The token-level entropy is a good signal for uncertainty. Since the CoT is long, aggregated entropy might work. If the entropy is below a threshold, we terminate the reasoning and force the model to generate a solution (appending \</think\>).
> - **Semantic similarity between adjacent steps**: When diving into the long CoT, we notice that the adjacent steps become more similar with a increasing number of steps. It shows the convergence of thinking. Inspired by this observation, we can measure the semantic similarity between adjacent steps. If the similarity is above a specified threshold, we can terminate the thinking and force the model to generate a solution.
>
> $~$
>
> > Q4. Selection Mechanism Robustness: "Did you notice any cases where these selection mechanisms failed or were biased by the presence of many partial solutions?"
>
> Yes, and we showed it in Table 1, i.e., the noisy predictions from early steps confuse the answer selection with majority voting, making "$H=16, m=1$" is slightly worse than "$H=1, m=1$" (65.3 vs 66.7 on average). With a PRM, "$H=16, m=1$" becomes better than "$H=1, m=1$" (61.4 vs 60.4). The reason is: majority voting equally treats all predictions, while PRM scores the prediction. With a simple early denoising step, i.e. "$H=-4$", we can significantly improve the accuracy by 6.6 for BoN and 4.6 for Maj.
>
> $~$
>
> > Q4. Selection Mechanism Robustness: "Could another solution be to weight answers by depth or confidence?"
>
> This is a great suggestion. From Figure 6, we observe that the overall accuracy is mostly proportional to the depth. It inspires us to design a linear depth-weighted selection for the $H$-dimension aggregation. For simplicity, we ignore the $m$ dimension here, i.e. setting $m=1$.
>
> Let LLM produce an answer $𝑎$ at depth $t$, where the reasoning depth $𝑡∈${$1,…,𝐻$}.
> For each produced instance of answer $𝑎$ at depth $𝑡$, we have a selector score $𝑠(a, t)$ (e.g., PRM score, or for majority voting set $𝑠(a, t)=1$) and a depth weight:
>
> $𝑤_{t}=\frac{t}{\sum_{k=1}^Hk}=\frac{2t}{H(H+1)}$​.
>
> This ensures:
> - deeper predictions receive higher weight,
> - early noisy predictions receive much lower weight,
> - weights sum to 1,
> - the mechanism is selector-agnostic (can plug into PRM or majority voting).
>
> We aggregate weighted scores for the same canonical answer across all occurrences (all $𝑡$ and trajectories):
>
> $S_{agg}(a) = \sum_{\text{all occurences of a}} w_t \cdot s(a, t)$
>
> Finally select:
>
> $\hat{𝑎} = argmax_a S_{agg}(a)$.
>
> As shwon in the following table, linear depth-weighted even performs better than the early denoising method ($H=-4$), and significantly improves the original $H=16$.
>
> | Metric | Method | H | m | MATH500 L5 | AIME24 | AIME25 | AIMO2 | GPQA | Avg. |
> | --- | --- | --- | --- | --- | --- | --- | --- | --- | --- |
> | Maj | Original | 1 | 1 | 95.5 | **76.7** | 60.0 | 50.0 | 51.5 | 66.7 |
> | Maj | Original | 16 | 1 | 94.0 | 73.3 | 60.0 | 50.0 | 49.0 | 65.3 |
> | Maj | Original | -4 | 1 | **96.3** | **76.7** | 63.3 | 60.0 | **53.0** | 69.9 |
> | Maj | Linear depth-weighted | 16 | 1 | 95.9 | **76.7** | **66.7** | **60.0** | 52.2 | **70.3** |
> | - | - | - | - | - | - | - | - | - | - |
> | BoN | Original | 1 | 1 | 90.3 | 63.3 | 53.3 | 40.0 | 55.1 | 60.4 |
> | BoN | Original | 16 | 1 | 90.3 | 70.0 | 53.3 | 40.0 | 53.5 | 61.4 |
> | BoN | Original | -4 | 1 | 93.3 | 73.3 | **60.0** | **60.0** | 53.5 | 68.0 |
> | BoN | Linear depth-weighted | 16 | 1 | **96.3** | **76.7** | **60.0** | **60.0** | **56.1** | **69.8** |
>
> ---
>
> $~$
>
> Please move to part 5

---

> > ### Author Response · Authors · 2025-11-27
> > **Response to Reviewer CQMk (part 5)**
> >
> > > Q5. Integration in APIs: "Do you have suggestions on how to approximate Fractured Sampling with commercial LLM APIs that don't allow control of mid-generation? Do you think the benefits of FS could be attained in such scenarios?"
> >
> > **Short Answer: Pure Fractured Sampling couldn't be applied with such APIs. But Fractured Sampling implemented in a multiturn conversation way could potentially work.**
> >
> > As we notice, only Gemini-2.5-Flash supports a visible thinking, while other frontier LLMs (Claude, GPT, Grok) don't even have a visible thinking (only the summarized version is shown). In additon, Gemini doesn't support a manipulation of the thinking process. Therefore, Fractured Sampling can't be seamlessly applied. It is the same challenge for prior works mentioned in the response to W1(1B).
> >
> > A potential workaround is implementing in a multiturn conversation way:
> > 1. For the first conversation turn, we prompt LLM to decompose the question into multiple sequential sub-questions or steps without answering it.
> > 2. For the second turn, we input the first sub-quesiton and prompt LLM to reason about it. This is considered as the first reasoning step.
> > 3. For the third turn, we prompt the LLM to answer the original question with the first reasoning step, and obtain the first prediction.
> > 4. Repeat the second and third steps till all sub-questions are resolved.
> >
> > Since each sub-question is simpler than the original question, the reaosning tokens should be fewer for each step. If the predictions converge, we can also stop early.
> >
> > ---
> >
> > $~$
> >
> > Thank you again for your thoughtful question and suggestion. Should you have any further questions, we are happy to assist.

---

### Official Review · Reviewer_6pRZ · 2025-11-01

**Soundness:** 2
**Presentation:** 2
**Contribution:** 2
**Rating:** 6
**Confidence:** 2

**Summary:**

The authors propose "Fractured Sampling" which samples solutions at multiple intermediate CoT truncation points (H), combined with multiple trajectories (n) and solutions per trajectory (m). The claim is this achieves better accuracy per token by exploiting the observation that truncated CoT often matches full CoT performance.

**Strengths:**

The results are nice.

**Weaknesses:**

LLMs generate unnecessary padding tokens after reaching correct answers. I feel like this is just rebranding early stopping + ensemble methods as a "unified framework" with three "orthogonal dimensions"? - which are actually just self-consistency (n), best-of-n (m), and early stopping (H)?

1. Figure 1 shows truncated CoT is better, but Table 1 shows H=16 degrades accuracy versus H=1 when using a PRM (61.4% vs 60.4%). They need to discard the first 11 positions as "noise" to make it work. If intermediate solutions are noisy, the whole premise fails?

2. Proposition 1 is using trivial inclusion-exclusion with fancy notation? I think the claim negative covariances help but Figure 3 shows mostly positive correlations (green cells dominate)?

3. This paper would be stronger with comparison to proper early stopping methods, speculative decoding, or any efficient inference techniques. They compare to naive "generate 32K tokens" baseline.

4. Scaling law C_H >= max{C_n, C_m} has little theoretical justification?Why should depth dominate? It would be nice to have an explanation here.

I feel like this method takes an observation (models overthink) and builds maybe an overcomplicated framework that performs worse than its own simplified version (H=-4 beats H=16)?


I am recommending a marginal accept, and can move my score to be higher if you address these questions. Thanks.

**Questions:**

Scaling law C_H >= max{C_n, C_m} has little theoretical justification?Why should depth dominate? It would be nice to have an explanation here.

---

> ### Author Response · Authors · 2025-11-21
> **Response to Reviewer 6pRZ (part 1)**
>
> Dear Reviewer 6pRZ,
>
> Thank you very much for your time, effort, and thorough review. We address your concerns below:
>
> $~$
>
> ---
>
> > Weakness. "Fractured Sampling is rebranding existing ideas (self-consistency, best-of-n, early stopping)"
>
> We respectfully disagree with this statement. We clarify that while our algorithm has conceptual similarities with some prior methods, ***the key novelty of Fractured Sampling is the H-dimension: sampling solutions at multiple intermediate reasoning depths within a single trajectory, which is not captured by prior inference-time techniques.***
>
> - Both self-consistency and Best-of-N uses multiple full trajectories ($n$-axis only).
> - Early stopping shortens a single trajectory, but does not sample or aggregate across multiple partial prefixes.
> - ***To the best of our knowledge, the m dimension (i.e. sample multiple solutions given the same reasoning) is also first studied.*** We found the $m$ dimension alone is more effective for the science domain (GPQA in Figure 4). When combining with the $n$ and $H$ dimensions (Figure 5 and Table 1), we achieved the best performance.
>
> Our method introduces temporal branching, enabling
> - aggregation of intermediate partial reasoning states;
> - temporal diversity unavailable to self-consistency;
> - three-way compute allocation under a unified scaling law. Empirically, this new dimension consistently yields steeper per-token scaling (Figure 4).
>
> Thus, Fractured Sampling is **NOT** a rebranding of existing axes but a strict generalization that introduces a fundamentally new mode of inference-time scaling.
>
> $~$
>
> > Weakness 1. "Figure 1 vs Table 1 — $H=16$ degrades accuracy with the PRM; early positions are ‘noise’ contradicting premise"
>
> Thank you for the careful reading. We want to politely clarify a small misunderstanding:
>
> - For PRM (Best-of-N), $H=16$ does improve accuracy over $H=1$ (61.4% vs. 60.4%).
> - The degradation occurs only under majority voting, where $H=16$ is slightly lower than $H=1$ (65.3% vs. 66.7%).
>
> We assume you was referring to the majority-voting case, and we address that concern below.
>
> 1. **Why can H=16 hurt majority voting even though it helps pass@k and PRM?**
>
> Majority voting aggregates raw predictions without any scoring. In the $H$-axis, early intermediate prefixes are indeed noisier (as shown in Figure 6). When all 16 intermediate predictions are included in a simple vote:
> - Early low-quality predictions contribute noise, diluting the signal from later high-quality depths.
> - Majority voting treats each prediction equally, so noisy early predictions dominate more than they should.
> - This effect is specific to majority voting; PRM-based selection and pass@k benefit from depth diversity because they are not forced to weight early predictions equally.
>
> This is precisely why the simple denoising strategy ($H = −4$) dramatically improves accuracy for both PRM and majority voting in Table 1.
>
> 2. **The observation from Figure 1 is based on a pre-condition: fixed token budget.**
>
> Truncated CoT performs much better than regular sampling for low token budget (<20K token/prompt), and is comparable to regular sampling with an increasing token budget (>20K token/prompt). Early steps are beneficial, because even noisy prefixes sometimes contain a correct premature reasoning.
>
> 3. **Does this contradict the premise of Fractured Sampling?**
>
> No. The core claim is that sampling across intermediate depths increases diversity and reduces joint failure probability, which is reflected in:
> - consistently higher pass@k for $H$-sampling,
> - higher PRM accuracy for $H=16$ vs $H=1$,
> - even stronger performance once early noisy predictions are removed ($H = −4$).
>
> The temporary degradation in majority voting is a property of the aggregation rule, not of the Fractured-Sampling principle.
>
> $~$
>
> > Weakness 2. "Proposition 1 seems trivial inclusion–exclusion"
>
> Our intention is not to claim a new combinatorial identity, but to derive a lower bound that explains why sampling across depths $H$ is theoretically beneficial. Prior sampling methods assume either independence (self-consistency) or identical distribution (Best-of-N). Proposition 1 explicitly:
>
> - models the dependency structure across intermediate reasoning states,
> - characterizes how negative covariance across depths amplifies success,
> - connects this structure to the empirical correlation matrices (Figure 3).
>
> While inclusion–exclusion is standard, the contribution is the application and interpretation: revealing that intermediate-step diversification systematically reduces joint failures even when marginals are similar.
>
> ---
>
> $~$
>
> Please move to part 2

---

> ### Author Response · Authors · 2025-11-21
> **Response to Reviewer 6pRZ (part 2)**
>
> > Weakness 2. "Negative covariances claimed, but Figure 3 shows mostly positive correlations"
>
> **Short Answer: Your observation that “positive correlations dominate” might be largely driven by the diagonal entries in Figure 3. These cells are always exactly 1 because each depth position is perfectly correlated with itself, so they visually bias the heatmap toward green. Actually, most cells show negative correlation.**
>
> To clarify the underlying structure, we computed the proportion of positive vs. negative correlations excluding diagonal elements in the following table. Across benchmarks, negative correlations are in fact more common, indicating that failures at different depths often do not co-occur. The only exception is MATH500, where the proportions are nearly equal. We believe this is because MATH500 is the easiest benchmark; the model’s reasoning path is more coherent and requires fewer reflection/correction steps, leading to more synchronized errors.
>
> Importantly, we do not expect correlations to be strongly negative across the board. Extremely low or consistently negative correlations would imply that reasoning steps are almost independent, which would indicate an incoherent or unstable reasoning trajectory. The observed moderate mixture of correlations is consistent with structured reasoning that still provides meaningful diversity across depths.
>
> |  | MATH500 | AIME24 | AIME25 | AIMO2 | GPQA |
> | --- | --- | --- | --- | --- | --- |
> | Positive correlation | 43.7% | 42.9% | 44.1% | 39.8% | 38.7% |
> | Negative correlation | 42.9% | 48.6% | 55.0% | 56.5% | 51.3% |
> | Neutral correlation | 13.4% | 8.6% | 0.9% | 3.7% | 10.1% |
>
> $~$
>
> > Weakness 3. "Comparison to early stopping, speculative decoding, efficient inference baseline missing"
>
> **Short Answer: We add two new baselines. Our early stopping method performs the best.**
>
> In Table 2, we only compare our proposed easly stop method to the vanilla sampling method (max_token=32K), because the vanilla method was considered as the upper bound. If our method could achieve a similar accuracy as the vanilla's with less number of tokens, it shows the effectiveness of our method.
>
> Thank you for this great suggestion. Here we further include two efficient sampling baselines for a thorough comparison. However, we exclude speculative decoding (SD), because our method is orthogonal to SD. SD applies a draft and target LLM to efficiently sample. Our Fractured Sampling can also be sampled with SD.
>
> The two new baselines are:
> - **Traditional early stop** with the same token budget as our method: When the number of reasoning tokens reaches 80% of the number of reasoning tokens from vanilla sampling, we force the LLM to generate the solution without further reasoning.
> - **LEASH** [1]: A recent proposed method (released on 6 Nov, 2025) that early stops the reasoning given the signal from entropy.
>
> **As shown in the following table for DS-R1-Qwen-1.5B, using the consistency across the $H$ dimension is a very effective signal for early stop. Although the new baselines save sampling tokens (LEASH saves the most), both of them couldn't match the accuracy of vanilla full sampling.**
>
> | Method | MATH500 | AIME25 | AIMO2 | GPQA | Avg accuracy | Avg #tokens (%) |
> | --- | --- | --- | --- | --- | --- | --- |
> | Vanilla full sampling | 70.8 | 27.5 | 15.0 | 34.1 | 36.9 | 100.0% |
> | Traditonal early stop | -1.0 | -2.2 | -5.0 | -0.9 | -2.3 | 80.0% |
> | LEASH | -4.6 | -3.5 | -3.6 | -1.6 | -3.3 | **71.2%** |
> | Our early stop | **+1.2** | **-0.0** | **+10.6** | **-0.3** | **+2.9** | 79.7% |
>
>
> [1] Logit–Entropy Adaptive Stopping Heuristic for Efficient Chain-of-Thought Reasoning, Mohammad Atif Quamar, Mohammad Areeb
>
> ---
>
> $~$
>
> Please move to part 3

---

> > ### Author Response · Authors · 2025-11-21
> > **Response to Reviewer 6pRZ (part 3)**
> >
> > > Weakness 4 and Question. "Scaling law $𝐶_H ≥$ max{$C_n, C_m$} lacks theoretical justification — why should depth dominate?"
> >
> > Thank you for raising this important question.
> >
> > We first note that existing scaling law for training [2,3] and test-time [4] commonly do not provide theoretical proofs for their empirical findings. Instead, the community has converged on the view that the value of scaling laws lies in the practical guidance they offer, not in formal derivations. Our goal is aligned with this tradition: to empirically characterize how allocating inference compute across different axes affects performance, and to offer actionable insight into how practitioners should budget tokens.
> >
> > **Why depth dominates in practice:**
> > Intuitively, sampling at multiple intermediate prefixes accesses temporally distinct latent states that capture the model’s evolving reasoning before it “commits” to a trajectory. These states often diverge earlier and more substantially than full trajectories sampled with different seeds (i.e. the $n$ dimension). This creates finer-grained, structurally diverse failure modes that the $n$- and $m$-axes cannot access:
> >
> > - $n$-axis: independent long-CoT trajectories often become semantically redundant due to inductive biases of RL-trained reasoning models.
> > - $m$-axis: samples share the exact same internal state, so diversity is limited.
> > - $H$-axis (ours): intermediate states are not yet converged, leading to decorrelated partial solutions and higher marginal gains per token.
> >
> > This explains the consistent empirical result (Figure 4) that $𝐶_H ≥$ max{$C_n, C_m$} across various models and tasks.
> >
> > In summary, our contribution is not a formal proof of a universal law (rarely exists in prior work either) but a practically valuable characterization showing that allocating compute to the $H$-dimension yields the steepest performance improvement per token. This practical guidance is precisely the purpose of test-time scaling laws.
> >
> > $~$
> >
> > [2] Scaling Laws for Neural Language Models, Jared Kaplan, Sam McCandlish, Tom Henighan ...
> >
> > [3] When scaling meets llm finetuning: The effect of data, model and finetuning method, Biao Zhang, Zhongtao Liu, Colin Cherry, Orhan Firat
> >
> > [4] s1: Simple test-time scaling, Niklas Muennighoff, Zitong Yang, Weijia Shi, ...
> >
> > ---
> >
> > $~$
> >
> > Please move to part 4

---

> ### Author Response · Authors · 2025-11-21
> **Response to Reviewer 6pRZ (part 4)**
>
> > Weakness. "Framework is overcomplicated"
>
> Thank you for raising this point. If our presentation gave the impression that the framework is complicated, we apologize. The underlying implementation is actually very simple: **Fractured Sampling only requires sampling solutions at multiple partial prefixes of a single CoT trajectory**. To help you easily understand Fractured Sampling, we:
> - Add Algorithm 1 (L162-L180) in the updated version;
> - Add a short illustrative GIF in the supplementary materials to clarify the procedure. You only need to download the  supplementary materials, open it with VSCode, and view README.
>
> $~$
>
> > Weakness. "simplified H=−4 variant outperforms H=16"
>
> Regarding the observation that the $H = −4$ setting outperforms $H = 16$,  as discussed in our response to Weakness 1, early prefixes can be noisy. And Maj and BoN are sensitive to this noise. Importantly, this does not undermine the contribution of Fractured Sampling. In Section 4.4, we also include a simple and practical truncated-depth variant—sampling solutions only after the first 6K reasoning tokens—which achieves strong performance while retaining the core benefit of depth-based sampling. This demonstrates that practitioners can easily adapt the depth-sampling interval to their selection mechanism.
>
> **The simplified $H = −4$ setting should be seen as a practical instantiation of our general framework rather than evidence against it.** The full Fractured Sampling formulation remains valuable because it provides a unified view of the $n/m/H$ allocation space and highlights why depth-diversified sampling offers consistently stronger per-token scaling across models and tasks.
>
> Suggested by Reviewer CQMk, we designed another more elegant approach to weight the predictions along the $H$ dimension. From Figure 6, we observe that the overall accuracy is mostly proportional to the depth. It inspires us to design a linear depth-weighted selection for the $H$-dimension aggregation. For simplicity, we ignore the $m$ dimension here, i.e. setting $m=1$.
>
> Let LLM produce an answer $𝑎$ at depth $t$, where the reasoning depth $𝑡∈${$1,…,𝐻$}.
> For each produced instance of answer $𝑎$ at depth $𝑡$, we have a selector score $𝑠(a, t)$ (e.g., PRM score, or for majority voting set $𝑠(a, t)=1$) and a depth weight:
>
> $𝑤_{t}=\frac{t}{\sum_{k=1}^Hk}=\frac{2t}{H(H+1)}$​.
>
> This ensures:
> - deeper predictions receive higher weight,
> - early noisy predictions receive much lower weight,
> - weights sum to 1,
> - the mechanism is selector-agnostic (can plug into PRM or majority voting).
>
> We aggregate weighted scores for the same canonical answer across all occurrences (all $𝑡$ and trajectories):
>
> $S_{agg}(a) = \sum_{\text{all occurences of a}} w_t \cdot s(a, t)$
>
> Finally select:
>
> $\hat{𝑎} = argmax_a S_{agg}(a)$.
>
> As shwon in the following table, linear depth-weighted even performs better than the early denoising method ($H=-4$), and significantly improves the original $H=16$.
>
> | Metric | Method | H | m | MATH500 L5 | AIME24 | AIME25 | AIMO2 | GPQA | Avg. |
> | --- | --- | --- | --- | --- | --- | --- | --- | --- | --- |
> | Maj | Original | 1 | 1 | 95.5 | **76.7** | 60.0 | 50.0 | 51.5 | 66.7 |
> | Maj | Original | 16 | 1 | 94.0 | 73.3 | 60.0 | 50.0 | 49.0 | 65.3 |
> | Maj | Original | -4 | 1 | **96.3** | **76.7** | 63.3 | 60.0 | **53.0** | 69.9 |
> | Maj | Linear depth-weighted | 16 | 1 | 95.9 | **76.7** | **66.7** | **60.0** | 52.2 | **70.3** |
> | - | - | - | - | - | - | - | - | - | - |
> | BoN | Original | 1 | 1 | 90.3 | 63.3 | 53.3 | 40.0 | 55.1 | 60.4 |
> | BoN | Original | 16 | 1 | 90.3 | 70.0 | 53.3 | 40.0 | 53.5 | 61.4 |
> | BoN | Original | -4 | 1 | 93.3 | 73.3 | **60.0** | **60.0** | 53.5 | 68.0 |
> | BoN | Linear depth-weighted | 16 | 1 | **96.3** | **76.7** | **60.0** | **60.0** | **56.1** | **69.8** |
>
>
> ---
>
> $~$
>
> Thank you again for your thoughtful question and suggestion. Should you have any further questions, we are happy to assist.

---

### Author Response · Authors · 2025-11-21
**Summary of paper update and rebuttal**

Dear AC and Reviewers,

To facilitate your discussion and decision, here we make a summary about the paper update and rebuttal.

---

## Paper update
we mainly make the following modification to the updated version:
-  **Notation and experiment details**: We standardize the notation in Section 2, and add more experimental details in Section 4 (in blue color) (**Reviewer BfLE**).
- **Scaling law w.r.t. latency**: We add a new scaling law figure with the x axis denoting the latency, Figure E.4 in appendix. ( **Reviewer CQMk and BfLE**)
- **Pseudocode**: We add Algorithm 1 in the main page for better demonstrating Fractured Sampling. (**Reviewer 6pRZ and BfLE**)
- **A short GIF for easy understanding of Fractured Sampling**: We also add a short GIF in the supplementary materials. You only need to open the README file with VSCode to view it. The frac_cot.gif file locates in the figs folder. (**Reviewer 6pRZ and BfLE**)

---

## Rebuttal summary

| Category | Main Weakness/Question Details (Reviewer) | Rebuttal Summary |
| --- | --- | --- |
| Novelty | Fractured Sampling is similar to self-consistency, best-of-n and early stopping (**6pRZ**) | Only conceptually similar, the $H$ and $m$ dimensions are first studied. |
| Contradiction between premise and observation | Noisy early predictions degrade BoN accuracy from Table 1, contradicting with Figure 1 where truncated CoT is better. (**6pRZ**) | Misunderstanding about Table 1 where $H=16$ degrade the Maj accuracy instead of the BoN accuracy. This is caused by the equally weighting from Maj. The Fractured-Sampling principle still hold. |
| Trivial proposition | Proposition 1 seems trivial inclusion–exclusion (**6pRZ**) | Our intention is not to claim a new combinatorial identity, but to derive a lower bound that explains why sampling across depths $H$ is theoretically beneficial |
| Contradiction between claim and observation | Negative covariances claimed, but Figure 3 shows mostly positive correlations (**6pRZ**) | The observation of “positive correlations dominate” is **misled** by the diagonal entries that always have a positive correlation.  The negative covariance actually dominates. |
| Baselines | Compare to early stopping, speculative decoding and other efficient inference baselines (**6pRZ**) | Two new baselines are added. Our early stopping algorithm obtains the best accuracy. |
| Theoretical justification | The scaling law lacks of theoretical justification (**6pRZ**) | We follow prior works to empirically show the scaling law, and explain why $H$ dimension helps. |
| Complicated framework | "Framework is overcomplicated" (**6pRZ**). Fractured Sampling requires many forward passes. (**CQMk**) | Fractured Sampling only requires one additional light-weight forward pass compared to self-consistency. It is very simple, and easy to implement. We add a pseudocode and a short GIF for better presentation. |
| Black-box APIs don't support | Black-box APIs don't allow mid-generation control. (**CQMk**) | As prior efficienct long-CoT works, we have the same assumption, i.e. accessing to the thinking content. We introduce a work-around with black-box API. |
| Lack of latency result | Only token budget result is shown. (**CQMk**) | Token as a budget estimation is more robust. We also add the latency result (Figure E.4) with the same findings. |
| Hyperparameter tuning | "Choosing $n$, $m$, $H$ requires trial-and-error." (**CQMk**) | The $H$ and $m$ dimensions are light weight, only minor extra tuning compared to self-consistency. |
| Limited discussion | Limited discussion of potential downsides, and how critical is the choice of PRM. (**CQMk**) | We honestly show the challenges of Fractured Sampling in the main pages, like noisy early predictions and the access requirement of thinking content. We add a new ablation study with a weaker PRM, showing a similar finding. |
| Generalization | Performance on reasoning-less tasks (**CQMk**) | New results on creative writing, showing a similar finding. |
| Presentation | "Pass@k is an evaluation metric”, notation need standarization, the base of log-scale, the model abbreviations. (**BfLE**) | We rephrase these parts, and add more details |
| Assumption | The scaling law assumes a unifrom cost. (**BfLE**) | Empirical results show a uniform cost. |
| Deeper investigation |  "At what prefix length correct answers typically emerge." (**BfLE**) | We add these new results. |
| Small benchmark | "Some benchmarks (AIME) are small." (**BfLE**) | We originally showed five benchmarks in the paper, and add a new coding benchmark (LiveCodeBench), showing a similar finding. |
| Interpretability | "Quality/interpretability of truncated reasoning not analyzed." (**BfLE**) | Our focus is on accuracy–cost trade-offs |
| Efficiency | "If the full trace must be generated before segmentation, how does Fractured Sampling save tokens?" (**BfLE**) | Fractured Sampling saves tokens, because it only requires one full trace. |

---

### Meta-Review · Area_Chair_tpdJ · 2026-01-07

**Summary:**

This submission explores a novel inference-time strategy to improve reasoning efficiency in large language models. However, the reviewers’ scores are low due to several critical issues. First, The paper suffers from unclear notation, lack of detailed pseudocode, and confusing explanations, making it difficult to understand and implement the proposed method. Second, the proposed method requires access to intermediate reasoning states and extensive hyper-parameter tuning, limiting its feasibility in black-box API settings. Additionally, the results are inconsistent across benchmarks, some of which are small in size, raising concerns about the generalizability of the findings.
Given these concerns and the overall low scores, I recommend rejecting this submission paper.

**Reviewer Concerns:**

The authors address Reviewer 6pRZ’s actionable concerns fairly well by adding missing baselines, clarifying the covariance/“noisy prefix” issue, and providing concrete fixes and extra results. However, the deeper skepticism (possible rebranding and the lack of a truly rigorous justification for the scaling-law claims) is only partially alleviated. Reviewer CQMk’s concerns about implementation, efficiency, and evaluation are mostly addressed through clearer procedural details and added latency/ablation evidence. However, the issue—practical viability under strict black-box APIs that don’t expose intermediate states—remains only partially resolved. The concerns of Reviewer BfLE are still outstanding.

**Reviewer Scores:**

I think the final scores of the reviewers are 6(Reviewer 6pRZ),4(Reviewer  CQMk),2(Reviewer BfLE). All the reviewers did not respond to the author. And part of concerns remain partially resolved.

---

### Decision · Program_Chairs · 2026-01-26

Reject